# Comprehensive analysis reveals potential therapeutic targets and an integrated risk stratification model for solitary fibrous tumors

Renjing Zhang[1,2,12], Yang Yang[3,12], Chunfang Hu[4,12], Mayan Huang[1,5,12], Wenjian Cen[1,2,12], Dongyi Ling[1,2], Yakang Long[1,2], Xin-Hua Yang[1,2], Boheng Xu[1,2], Junling Peng[1,2], Sujie Wang[1,2], Weijie Zhu[1,2], Mingbiao Wei[1,2], Jiaojiao Yang[1,2], Yuxia Xu[1,2], Xu Zhang[1,2], Jiangjun Ma[1,2], Fang Wang [1,2], Hongtu Zhang[4], Peiqing Ma[4], Xiaojun Zhu[1,6], Guohui Song[1,6], Li-Yue Sun [7], De-Shen Wang [1,8], Feng-Hua Wang[1,8], Yu-Hong Li[1,8], Sandro Santagata [9], Qin Li [10,11,13] ✉, Yan-Fen Feng [1,5,13] ✉ & Ziming Du [1,2,13] ✉

Solitary fibrous tumors (SFTs) are rare mesenchymal tumors with unpredictable evolution and with a recurrence or metastasis rate of 10-40%. Current medical treatments for relapsed SFTs remain ineffective. Here, we identify potential therapeutic targets and risk factors, including *IDH1* p.R132S, high PD-L1 expression, and predominant macrophage infiltration, suggesting the potential benefits of combinational immune therapy and targeted therapy for SFTs. An integrated risk model incorporating mitotic count, density of Ki-67+ cells and CD163+ cells, *MTOR* mutation is developed, applying a discovery cohort of 101 primary non-CNS patients with negative tumor margins (NTM) and validated in three independent cohorts of 210 SFTs with the same criteria, and in 36 primary CNS SFTs with NTM. Compared with the existing models, our model shows significantly improved efficacy in identifying high-risk primary non-CNS and CNS SFTs with NTM for tumor progression. Our findings hold promise for advancing therapeutic strategies and refining risk prediction in SFTs.

Solitary fibrous tumors (SFTs), characterized by an *NAB2-STAT6* gene fusion[1,2], are rare fibroblastic mesenchymal tumors that can occur at any anatomic location, especially pleural, meningeal, or extrapleural sites[3,4]. According to the fifth edition of the WHO (World Health Organization) classification of soft tissue and bone tumor[5], non-central nervous system (non-CNS) SFTs were classified as benign SFT (intermediate category, locally aggressive), SFT NOS (Not Otherwise Specified, intermediate category, rarely metastasizing), and malignant SFT. In addition, the CNS SFTs were classified as WHO grade 1 (<5 mitoses/ 10 HPF), WHO grade 2 (≥5 mitoses/10 HPF without necrosis), WHO

grade 3 (≥5 mitoses/10 HPF with necrosis) according to the fifth edition of the WHO classification of CNS tumors[6]. Despite an indolent course, aggressive behavior in the form of local recurrence or distant metastasis may still occur in 10–40% of the SFT patients[7,8]. Since most SFTs are insensitive to conventional chemotherapy, and there are no standard guidelines for the systemic treatment of SFTs, the prognosis of patients with recurrent or metastatic SFTs remains poor[9].

SFTs comprise a histological spectrum and have an unpredictable evolution, and some SFTs can still progress despite having a clearly benign appearance. To improve prognostic accuracy, several risk

stratification models that composed of multiple prognostic factors, mainly clinical and histopathological variables, have been proposed to predict the individual risk of recurrence/metastasis[7,10–14]. Unfortunately, their predictive accuracy remains suboptimal[14–16], necessitating the inclusion of additional factors such as molecular and immune infiltrate analysis[17].

To address these challenges, in this work, we aim to characterize the genomic landscape and immune infiltrate in SFT tissues to explore potential therapeutic targets, and develop an improved risk stratification model. Notably, we identified an actionable mutation, *IDH1* p.R132S, in 6.9% of SFTs, and observed high PD-L1 expression in tumor or immune cells in 24.4% of cases, suggesting potential benefits of combination of immune therapy and targeted therapy. Moreover, we developed a three-tiered integrated risk stratification model incorporating mitotic count, density of Ki-67+ and CD163+ cells, and *MTOR* mutation to more accurately identify SFT patients of primary non-CNS and CNS SFTs with negative tumor margins (NTM) at high risk for tumor progression (Fig. 1). Our findings hold promise for advancing therapeutic strategies and refining risk prediction in SFTs.

## Results
### Clinicopathological and histopathological finding

A total number of 408 cases of SFTs were utilized as one discovery cohort (SYSUCC cohort, *n* = 131), and three independent validation

cohorts (FAHSYSU cohort, *n* = 115; CHCAMS cohort 1, *n* = 101; CHCAMS cohort 2, *n* = 61) (Table 1, Supplementary Data 1). Detection of the *NAB2–STAT6* fusion gene at the DNA level with fluorescent in situ hybridization, DNA-based PCR or sequencing is impossible because of the small size of the inverted sequence, proximity of the *NAB2* and *STAT6* loci, and the diversity of possible breakpoints in fusion transcripts[9]. For routine diagnosis, STAT6 is a robust immunohistochemical surrogate marker of all *NAB2–STAT6* fusion with excellent sensitivity (86-98%) in SFTs[18–21]. In this study, we found that STAT6 was positive in 91.60% (120/131) of SYSUCC cohort, 98.26% (113/115) of FAHSYSU cohort, 96.04% (97/101) of CHCAMS cohort 1, and 98.36% (60/61) of CHCAMS cohort 2 (Table 1, Supplementary Table 1, Supplementary Fig. 1).

Under further investigation, among the 11 cases with STAT6 IHC negative in SYSUCC cohort, 6 cases SFTs were resected within 5–10 years, while 5 cases SFTs were resected more than 10 years (Supplementary Table 2). Hence the loss of staining was consistent with decreased antigenicity in old tissue blocks as described previously[18]. RT-PCR (Reverse Transcription Polymerase Chain Reaction) was performed to detect eight most common types of *NAB2-STAT6* fusion in mRNA level in these 11 SFT samples. While 2 case were failed with RNA extraction, we found 4 cases SFTs with *NAB2* exon 4 - *STAT6* exon 2 fusion, 1 case with *NAB2* exon 2 - *STAT6* exon 5 fusion, and 4 cases without any of these 8 common types of *NAB2-STAT6* fusion detected

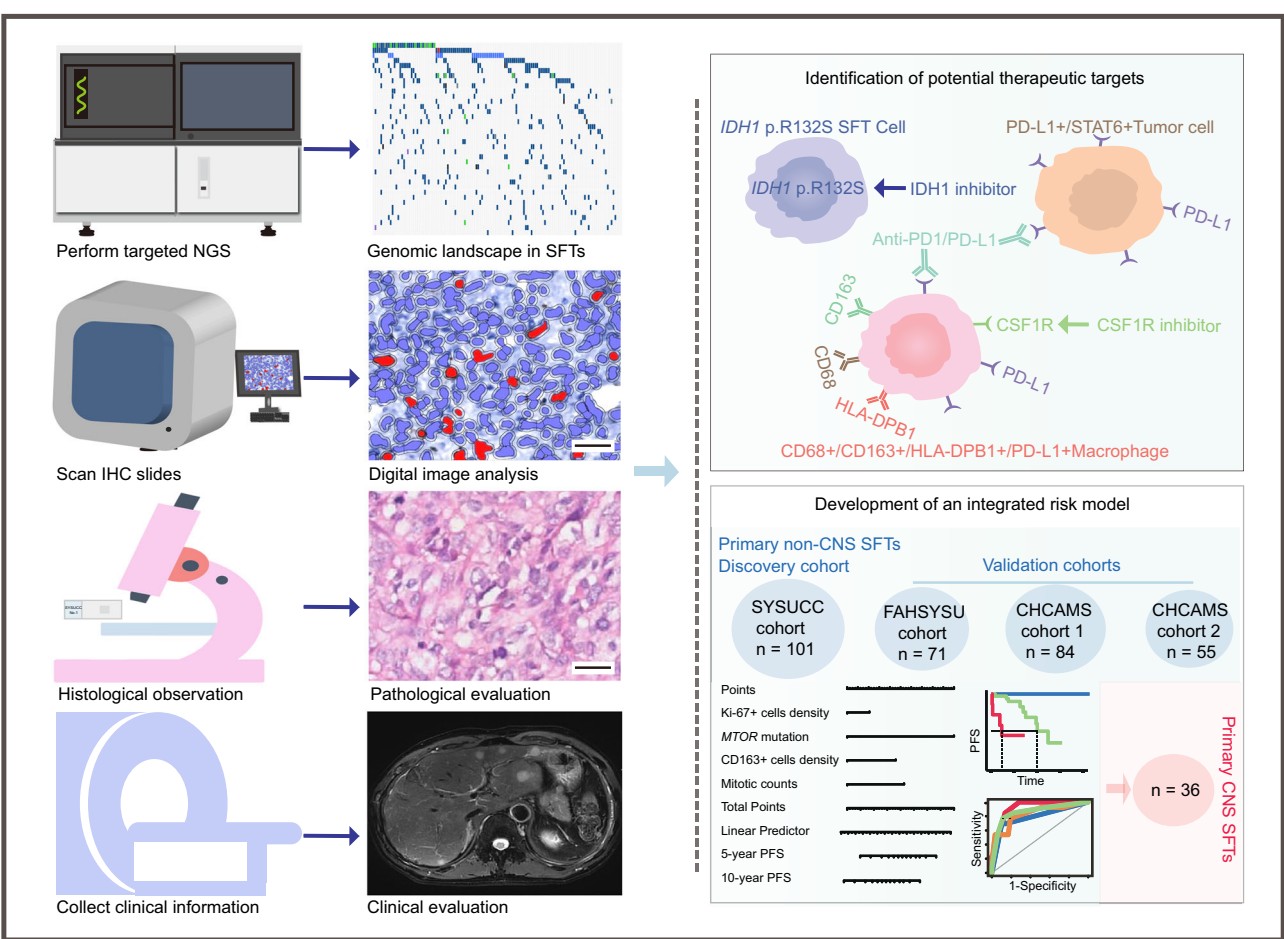

**Fig. 1 | Schematic outline of the study.** Genomic landscape, immune infiltrate and clinicopathological parameters were investigated retrospectively in SFTs. Potential drug therapeutic targets were identified, including *IDH1* p.R132S, PD-L1, and macrophages. An integrated risk model for primary non-CNS SFTs with NTM incorporated that information above was developed using a discovery cohort of SYSUCC (*n* = 101) and three validation cohorts of FAHSYSU (*n* = 71) and CHCAMS1 (*n* = 84) and CHCAMS2 (*n* = 55). In addition, the integrated model was validated for primary non-CNS SFTs with NTM (*n* = 36). (scale bars: 50 μm). NGS Next Generation Sequencing, IHC Immunohistochemistry, SFTs Solitary Fibrous Tumors, SYSUCC Sun Yat-Sen University Cancer Center, FAHSYSU The First Affiliated Hospital of Sun Yat-sen University, CHCAMS Cancer Hospital Chinese Academy of Medical Sciences.

**Table 1 | Patients demographics in SYSUCC cohort and three validation cohorts (FAHSYSU, CHCAMS 1, and CHCAMS 2)**

| Characteristics | Total (n = 408) | SYSUCC cohort (n = 131) | FAHSYSU cohort (n = 115) | CHCAMS cohort 1 (n = 101) | CHCAMS cohort 2 (n = 61) |
|---|---|---|---|---|---|
| **Sex** | | | | | |
| Male | 195 (47.79%) | 67 (51.15%) | 57 (49.57%) | 41 (40.59%) | 30 (49.18%) |
| Female | 213 (52.21%) | 64 (48.85%) | 58 (50.43%) | 60 (59.41%) | 31 (50.82%) |
| **Age at diagnosis (years)** | | | | | |
| Median | 51 (9.00–87.00) | 51 (9.00–87.00) | 45 (15.00–80.00) | 54 (23.00–75.00) | 55 (21.00–75.00) |
| ≥55 | 164 (40.20%) | 52 (39.69%) | 31 (26.96%) | 49 (48.51%) | 32 (52.46%) |
| <55 | 244 (59.80%) | 79 (60.31%) | 84 (73.04%) | 52 (51.49%) | 29 (47.54%) |
| **STAT6 expression** | | | | | |
| Score 0/1 | 18 (4.41%) | 11 (8.40%) | 2 (1.74%) | 4 (3.96%) | 1 (1.64%) |
| Score 2/3 | 390 (95.59%) | 120 (91.60%) | 113 (98.26%) | 97 (96.04%) | 60 (98.36%) |
| **Tumor sites** | | | | | |
| Intra-thoracica | 186 (45.59%) | 62 (47.33%) | 17 (14.78%) | 68 (67.33%) | 39 (63.93%) |
| Head and neck | 26 (6.37%) | 7 (5.34%) | 16 (13.91%) | 1 (0.99%) | 2 (3.28%) |
| Trunk | 53 (12.99%) | 15 (11.45%) | 24 (20.87%) | 8 (7.92%) | 6 (9.84%) |
| Extremity | 15 (3.68%) | 3 (2.29%) | 10 (8.70%) | 1 (0.99%) | 1 (1.64%) |
| Intra-abdominal | 70 (17.16%) | 25 (19.08%) | 21 (18.26%) | 12 (11.88%) | 12 (19.67%) |
| CNS | 58 (14.22%) | 19 (14.50%) | 27 (23.48%) | 11 (10.89%) | 1 (1.64%) |
| **Specimen type** | | | | | |
| Primary | 356 (87.25%) | 117 (89.31%) | 94 (81.74%) | 90 (89.11%) | 55 (90.16%) |
| Recurrence | 42 (10.29%) | 11 (8.40%) | 16 (13.91%) | 10 (9.90%) | 5 (8.20%) |
| Metastasis | 10 (2.45%) | 3 (2.29%) | 5 (4.35%) | 1 (0.99%) | 1 (1.64%) |
| **Margins** | | | | | |
| R0[a] | 391 (95.83%) | 127 (96.95%) | 106 (92.17%) | 98 (97.03%) | 60 (98.36%) |
| R1[b] | 7 (1.72%) | 2 (1.53%) | 1 (0.87%) | 3 (2.97%) | 1 (1.64%) |
| R2[c] | 10 (2.45%) | 2 (1.53%) | 8 (6.96%) | 0 (0.00%) | 0 (0.00%) |
| **Radiotherapy** | | | | | |
| No | 377 (92.40%) | 116 (88.55%) | 106 (92.17%) | 96 (95.05%) | 59 (96.72%) |
| Yes | 31 (7.60%) | 15 (11.45%) | 9 (7.83%) | 5 (4.95%) | 2 (3.28%) |
| **Chemotherapy** | | | | | |
| No | 389 (95.34%) | 124 (94.66%) | 109 (94.78%) | 97 (96.04%) | 59 (96.72%) |
| Yes | 19 (4.66%) | 7 (5.34%) | 6 (5.22%) | 4 (3.96%) | 2 (3.28%) |
| **Status** | | | | | |
| Recurrence | 47 (11.52%) | 16 (12.21%) | 16 (13.91%) | 8 (7.92%) | 7 (11.48%) |
| Metastasis | 29 (7.11%) | 8 (6.11%) | 9 (7.83%) | 8 (7.92%) | 4 (6.56%) |
| Died | 11 (2.70%) | 5 (3.82%) | 4 (3.48%) | 0 (0.00%) | 2 (3.28%) |
| Progression-free | 325 (79.66%) | 103 (78.63%) | 88 (76.52%) | 85 (84.16%) | 49 (80.33%) |
| **Metastatic sites** | | | | | |
| Pleura/lung | 7 (1.72%) | 1 (0.76%) | 1 (0.87%) | 3 (2.97%) | 2 (3.28%) |
| Liver | 5 (1.23%) | 2 (1.53%) | 1 (0.87%) | 1 (0.99%) | 1 (1.64%) |
| Bone | 6 (1.47%) | 1 (0.76%) | 3 (2.61%) | 2 (1.98%) | 0 (0.00%) |
| Peritoneum/ intraperitoneal | 4 (0.98%) | 0 (0.00%) | 4 (3.48%) | 0 (0.00%) | 0 (0.00%) |
| Lymphonodus | 2 (0.49%) | 2 (1.53%) | 0 (0.00%) | 0 (0.00%) | 0 (0.00%) |
| Brain | 5 (1.23%) | 2 (1.53%) | 0 (0.00%) | 2 (1.98%) | 1 (1.64%) |
| Follow-up time (months) | 36.90 (0.00–171.30) | 41.10 (0.80–171.30) | 29.10 (0.00–121.20) | 32.20 (0.00–73.90) | 62.30 (0.30–121.10) |

*CNS* Central Nervous System.
[a]R0 Microscopic Complete.
[b]R1 Microscopic Incomplete.
[c]R2 Macroscopic Incomplete.

(Supplementary Table 2). Taken together, 95.42% (125/131) of SFTs in SYSUCC cohort had either STAT6 IHC positive (n = 120) or *NAB2-STAT6* fusion detected (n = 5). The remaining 6 cases were re-reviewed by an experienced soft tissue pathologist, and were confirmed as SFT histologically.

Among all 408 SFTs in these four cohorts, 85.78% (350/408) were non-CNS SFTs, whose PFS was significantly longer than that of CNS SFTs (14.22%, 58/408) (p = 2.49E−07, Table 1, Supplementary Fig. 2a), which may be due to different biological behavior and surgical difficulty of non-CNS SFTs and CNS SFTs. In addition, as expected, both the relapsed SFTs (12.75%, 52/408) and SFTs with tumor margin positive (4.17%, 17/408) had shorter PFS than that of primary SFTs and SFTs with NTM respectively (p = 2.00E−16 and p = 2.40E−07, Table 1, Supplementary Fig. 2b–c). Notably the SFTs patients with adjuvant

radiotherapy or chemotherapy even had shorter PFS than that of SFTs patients without adjuvant treatment ($p$ = 3.20E−05 and $p$ = 2.71E−07, respectively, Table 1, Supplementary Fig. 2d–e). It may be attributed to that the SFTs patients selected for adjuvant treatment (usually malignant SFTs, or relapsed SFTs, or margin positive SFTs) were with worse condition and efficacy of adjuvant treatment on them was limited. In addition, tumor sites were not associated with PFS of primary non-CNS SFTs (Supplementary Fig. 2f)

Hence, considering the findings above, we only used primary non-CNS SFTs with NTM in SYSUCC cohort ($n$ = 101) for the integrated risk model generation, which was then validated in three cohorts with the same criteria (Table 2). In addition, the integrated risk model was also tested in both primary CNS SFTs with NTM (Supplementary Table 3) and the relapsed (recurrent and metastatic) non-CNS SFTs with NTM (Supplementary Table 4) from all four cohorts. The median follow-up time was 50.40 (1.50–171.30) months in SYSUCC cohort, 32.30 (0.00–121.20) months in FAHSYSU cohort, 36.00 (0.00–73.90) months in CHCAMS cohorts 1, and 72.60 (0.30–121.10) months in CHCAMS cohort 2, respectively (Table 2). Univariable survival analysis revealed that the variables of mitotic count, WHO classification were associated with PFS in all four cohorts, but the prognostic value of other variables, including patient sex, age at diagnosis, tumor size, nuclear pleomorphism, cellularity, necrosis, and adjuvant treatment, was not consistent across four cohorts (Table 2). In the cohort of primary CNS SFTs with NTM ($n$ = 36), the variables of mitotic count, necrosis, WHO grade were associated with PFS (Supplementary Table 3). However, in the cohort of relapsed non-CNS SFTs cohort with NTM ($n$ = 31), no significant association was observed between any variables and patients PFS (Supplementary Table 4), and it may be due to complexity of relapsed SFTs cases.

## Genomic landscape and clinical actionable gene alterations in SFTs

To explore the genomic landscape of SFTs, targeted NGS was performed to simultaneously detect 1021 cancer-related genes[22] in the discovery cohort ($n$ = 131). The most frequently altered genes in SFTs included *ZFHX3* (25%), *MLL3* (21%), *TERT* (19%), *SLX4* (17%), *FAT1* (16%), *MLL2* (16%), *ARID1B* (14%), *CDH23* (13%), *NOTCH1* (9%) (Fig. 2a, Supplementary Data 2). We showed that 17.6% (23/131) of SFTs have clinically actionable genomic alterations, which are potential targets mainly for isocitrate dehydrogenase 1 (IDH1) inhibitor, poly [ADP-ribose] polymerase (PARP) inhibitor and mammalian target of rapamycin (MTOR) inhibitor (Fig. 2b). Notably, we identified *IDH1* p.R132S in 6.9% (9/131) of SFTs by NGS which was then verified by Sanger Sequencing (Supplementary Table 5, Supplementary Fig. 3a–b). *IDH1* p.R132S mutation was highly enriched in SFT cases with WHO classification of malignant, high nuclear pleomorphism and cellularity (Supplementary Fig. 3c–e), but was not associated with patient PFS (Supplementary Fig. 3f).

Alterations of *MTOR* (Fig. 2c, $p$ = 3.00E−05), *TP53* (Fig. 2d, $p$ = 0.045), and *ERCC5* (Supplementary Fig. 3g, $p$ = 0.0024) were associated with shorter PFS in SFT patients. Alterations in the NOTCH signaling pathway, including *NOTCH1, NOTCH2, NOTCH3* and *CREBBP* were related to the PFS of the patients (Supplementary Fig. 3h). The *TERT* promoter region mutations were found in 19.08% (25/131) SFTs, but it was not associated with PFS in SFT patients in this cohort (Fig. 2e). However, *TERT* promoter region mutations were associated with malignant histology, necrosis, large tumor size, and older age (Supplementary Fig. 3i–l). *TP53* mutations were more common in malignant SFTs, male SFTs, and SFTs with high mitotic counts (Supplementary Fig. 3m–o). In addition, *MTOR* mutations were detected in 3.05% (4/131) of SFTs in discovery cohort, in 8.70% (10/115) of SFTs in FAHSYSU cohort, 3.96% (4/101) of SFTs in CHCAMS cohort 1, and 4.92% (3/61) of SFTs in CHCAMS cohort 2 (Supplementary Table 6, Supplementary Data 1, Supplementary Data 3).

## Characterization of the immune infiltrate in SFTs

In SFTs, tumor-infiltrating immune cells were predominantly macrophages (Fig. 3a). The median cell densities of CD68+ macrophages, CD163+ macrophage and HLA-DPB1+ cells in SFTs were 42.70 (Inter quartile range, IQR: 16.60–116.10) cells/mm$^2$, 549.50 (IQR: 331.70–1011.90) cells/mm$^2$ and 2029.20 (IQR: 1193.60–4083.70) cells/mm$^2$ respectively, whereas the median cell densities of CD3+ T cells, CD4+ cells, CD8+ T cells, FOXP3+ Treg cells, CD11c+ dendritic cells, and CD20+ B cells were 121.10 (IQR: 65.50–277.40) cells/mm$^2$, 388.70 (IQR: 163.60–932.50) cells/mm$^2$, 20.60 (IQR: 10.00–72.40) cells/mm$^2$, 13.90 (IQR: 6.10–34.50) cells/mm$^2$, 33.10 (IQR: 9.60–178.70) cells/mm$^2$ and 49.80 (IQR: 23.70–92.30) cells/mm$^2$ respectively (Fig. 3a). Notably, the high cell densities of CD68+ macrophages, CD163+ macrophages and HLA-DPB1+ cells in SFTs were significantly correlated with shorter PFS (Fig. 3b–d, Supplementary Table 7). However, neither CD3+ nor CD8+ cell density was associated with PFS (Supplementary Table 7). In addition, interestingly, the high cell densities of CD20+ B cells in SFTs were significantly correlated with longer PFS (Supplementary Table 7). We also found that the high density of Ki-67 positive cells correlated with short PFS in the SYSUCC cohort (Supplementary Table 7).

Because PD-L1 expression is the most important factor for predicting response to anti-PD-1 blockade therapy[23], we investigated PD-L1 expression in SFTs, and found 13.74% (18/131) of SFTs had high PD-L1 expression in tumor cells and 10.69% (14/131) of SFTs had high PD-L1 expression in immune cells, especially in CD68+/ HLA-DPB1+/CD163+ macrophages (Fig. 3e, Supplementary Fig. 4a). High PD-L1+ expression in immune cells was significantly associated with WHO classification, nuclear pleomorphism, and tumor site, but was not significantly associated with PFS (Supplementary Fig. 4b–e). Increased density of CD20+ B cells was enriched in SFTs patients with tumor expressing PD-L1 and was associated with longer PFS (Supplementary Fig. 4f–g). In addition, we found that two SFT cases showed both *IDH1* p.R132S mutation and high PD-L1 expression (Supplementary Fig. 4h).

## Development and validation of an integrated risk model in SFTs

For risk modeling, as mentioned above, we only used primary non-CNS SFTs with NTM in SYSUCC cohort ($n$ = 101) for the integrated risk model generation, which was then validated in three cohorts (FAHSYSU, $n$ = 71; CHCAMS cohort 1, $n$ = 84; CHCAMS cohort 2, $n$ = 55) with the same criteria (Table 2).

First, we used LASSO regression and random survival forest to evaluate and rank the variables from clinical and histopathological factors, immunohistochemical factors and molecular factors respectively in SYSUCC discovery cohort, and then the common variables predicted by these two methods were included in the final nomogram integrated risk model (Supplementary Fig. 5a–d). Finally, the four variables of mitotic count, density of Ki-67+ and CD163+ cells, and *MTOR* mutation were included in the integrated model (Fig. 4a). The total score was calculated for each patient using nomogram integrated risk model. Patients were stratified into three risk groups: low-risk (total score = 0), intermediate-risk (0 < total score ≤ 3.37), and high-risk (total score > 3.37) (Table 3).

In discovery SYSUCC cohort (n = 101), the integrated risk model stratified SFTs into low-risk group, intermediate-risk group, and high-risk group at 62.38% (63/101), 30.69% (31/101) and 6.93% (7/101) respectively, and the corresponding 3-year PFS was 100%, 89% and 43% ($p$ = 7.49E−12), respectively (Fig. 4b, Supplementary Table 8). The 3-year PFS of low-risk group, intermediate-risk group, and high-risk group was 100%, 95%, and 48% in FAHSYSU cohort ($p$ = 4.81E−05) (Fig. 4c, Supplementary Table 8), and 100%, 75% and 76% in CHCAMS cohort 1 ($p$ = 0.039) (Fig. 4d, Supplementary Table 8). In addition, the 5-year PFS of low-risk group, intermediate-risk group, and high-risk group was 100%, 83%, and 18% in CHCAMS cohort 2 ($p$ = 3.30E−06) (Fig. 4e, Supplementary Table 8). Moreover, the integrated model was validated in primary CNS SFTs with NTM ($n$ = 36) (Supplementary

**Table 2 | Univariate analysis for PFS in SYSUCC cohort, FAHSYSU cohort, CHCAMS cohort 1 and CHCAMS cohort 2 of primary non-CNS SFTs with negative tumor margins**

| Clinicopathologic feature | Total number (%) | SYSUCC cohort (n = 101) | | | FAHSYSU cohort (n = 71) | | | CHCAMS cohort 1 (n = 84) | | | CHCAMS cohort2 (n = 55) | | |
|---|---|---|---|---|---|---|---|---|---|---|---|---|---|
| | | Number (%) | Univariate analysis HR (95%CI) | P value | Number (%) | Univariate analysis HR (95%CI) | P value | Number (%) | Univariate analysis HR (95%CI) | P value | Number (%) | Univariate analysis HR (95%CI) | P value |
| **Sex** | | | | 0.041 | | | 0.360 | | | 0.622 | | | 0.118 |
| Male | 144 (46.30%) | 49 (48.51%) | 3.50 (1.18–10.40) | | 33 (46.48%) | 1.83 (0.48–6.91) | | 33 (39.29%) | 0.65 (0.13–3.29) | | 29 (52.73%) | 0.29 (0.07–1.31) | |
| Female | 167 (53.70%) | 52 (51.49%) | 1.00 | | 38 (53.52%) | 1.00 | | 51 (60.71%) | 1 | | 26 (47.27%) | 1.00 | |
| **Age at diagnosis (years)** | | | | 0.006 | | | 0.302 | | | 0.429 | | | 0.202 |
| ≥55 | 124 (39.87%) | 42 (41.58%) | 5.07 (1.67–15.40) | | 15 (21.13%) | 2.19 (0.28–16.85) | | 41 (48.81%) | 1.96 (0.39–9.70) | | 26 (47.27%) | 2.78 (0.63–12.25) | |
| <55 | 187 (60.13%) | 59 (58.42%) | 1.00 | | 56 (78.87%) | 1.00 | | 43 (51.19%) | 1 | | 29 (52.73%) | 1.00 | |
| **Tumor size (cm)** | | | | 0.408 | | | 0.012 | | | 0.004 | | | 0.310 |
| 15≤D | 56 (18.00%) | 24 (23.76%) | 2.37 (0.64–8.81) | | 6 (8.45%) | 3.46 (0.32–37.39) | | 12 (14.29%) | 11.48 (1.55–84.98) | | 14 (25.45%) | 4.71 (0.44–50.28) | |
| 10≤D<15 | 52 (16.72%) | 20 (19.80%) | 1.03 (0.17–6.19) | | 9 (12.68%) | 3.89 (0.54–28.32) | | 7 (8.33%) | 0.30 (0.00–49.41) | | 16 (29.09%) | 5.94 (0.60–58.84) | |
| 5≤D<10 | 106 (34.08%) | 32 (31.68%) | 0.71 (0.12–4.11) | | 28 (39.44%) | 0.31 (0.04–2.19) | | 32 (38.10%) | 1.21 (0.07–19.63) | | 14 (25.45%) | 5.29 (0.10–289.31) | |
| D<5 | 97 (31.19%) | 25 (24.75%) | 1.00 | | 28 (39.44%) | 1.00 | | 33 (39.29%) | 1 | | 11 (20.00%) | 1.00 | |
| **Cellularity** | | | | 0.001 | | | 0.373 | | | 0.001 | | | 0.001 |
| High | 37 (11.90%) | 9 (8.91%) | 8.37 (0.89–78.98) | | 15 (21.13%) | 2.80 (0.57–13.67) | | 6 (7.14%) | 0.32 (0.00–11.38) | | 7 (12.73%) | 718.79 (30.63–16865.81) | |
| Moderate | 50 (16.08%) | 14 (13.86%) | 4.61 (0.94–22.73) | | 13 (18.31%) | 1.95 (0.26–14.54) | | 9 (10.71%) | 6.87 (0.62–75.97) | | 14 (25.45%) | 17.63 (2.25–138.03) | |
| Low | 224 (72.03%) | 78 (77.23%) | 1.00 | | 43 (60.56%) | 1.00 | | 69 (82.14%) | 1 | | 34 (61.82%) | 1.00 | |
| **Necrosis** | | | | 0.000 | | | 0.403 | | | 0.643 | | | 0.046 |
| Present | 36 (11.58%) | 14 (13.86%) | 6.47 (1.19–35.15) | | 8 (11.27%) | 1.92 (0.27–13.46) | | 9 (10.71%) | 1.65 (0.13–21.79) | | 5 (9.09%) | 4.57 (0.30–69.65) | |
| Absent | 275 (88.42%) | 87 (86.14%) | 1.00 | | 63 (88.73%) | 1.00 | | 75 (89.29%) | 1 | | 50 (90.91%) | 1.00 | |
| **Nuclear pleomorphism** | | | | 0.020 | | | 0.072 | | | 0.015 | | | 1.79E−11 |
| High | 27 (8.68%) | 10 (9.90%) | 4.39 (0.91–21.18) | | 7 (9.86%) | 69.40 (2.65–1817.03) | | 8 (9.52%) | 15.22 (0.45–520.38) | | 2 (3.64%) | 50.50 (0.00–4911437.61) | |
| Moderate | 96 (30.87%) | 23 (22.77%) | 2.96 (0.60–14.55) | | 34 (47.89%) | 4.97 (1.07–23.20) | | 25 (29.76%) | 8.00 (0.89–72.16) | | 14 (25.45%) | 6.63 (1.00–44.17) | |
| Low | 188 (60.45%) | 68 (67.33%) | 1.00 | | 30 (42.25%) | 1.00 | | 51 (60.71%) | 1 | | 39 (70.91%) | 1.00 | |
| **Mitoses / 10 HPF** | | | | 1.35E−07 | | | 0.001 | | | 0.000 | | | 0.001 |
| ≥4 | 73 (23.47%) | 22 (21.78%) | 13.94 (3.52–55.28) | | 23 (32.39%) | 9.23 (2.14–39.88) | | 15 (17.86%) | 18.46 (2.62–130.30) | | 13 (23.64%) | 10.28 (1.59–66.55) | |
| <4 | 238 (76.53%) | 79 (78.22%) | 1.00 | | 48 (67.61%) | 1.00 | | 69 (82.14%) | 1 | | 42 (76.36%) | 1.00 | |
| **WHO classification** | | | | 3.28E−06 | | | 0.019 | | | 1.92E−05 | | | 0.000 |
| Malignant | 53 (17.04%) | 18 (17.82%) | 10.41 (2.51–43.16) | | 14 (19.72%) | 5.97 (1.12–31.92) | | 11 (13.10%) | 27.19 (2.99–247.70) | | 10 (18.18%) | 12.95 (1.73–96.81) | |
| Rarely metastasizing, NOS | 15 (4.82%) | 3 (2.97%) | 0.35 (0.00–42.70) | | 7 (9.86%) | 2.25 (0.12–41.74) | | 2 (2.38%) | 0.36 (0.00–247220.77) | | 3 (5.45%) | 0.34 (0.00–84.65) | |
| Locally aggressive, Benign | 243 (78.14%) | 80 (79.21%) | 1.00 | | 50 (70.42%) | 1.00 | | 71 (84.52%) | 1 | | 42 (76.36%) | 1.00 | |
| **Tumor sites** | | | | 0.148 | | | 0.634 | | | 1.92E−05 | | | 0.384 |
| Intra-thoracica | 177 (56.91%) | 60 (59.41%) | 1.00 | | 16 (22.54%) | 1.00 | | 64 (76.19%) | 1 | | 37 (67.27%) | 1.00 | |
| Head and neck | 22 (7.07%) | 6 (5.94%) | 5.28 (0.28–98.90) | | 13 (18.31%) | 5.59 (0.34–92.75) | | 1 (1.19%) | – | | 2 (3.64%) | 5.65 (0.07–438.93) | |
| Trunk | 36 (11.58%) | 9 (8.91%) | 2.10 (0.28–15.76) | | 18 (25.35%) | 5.57 (0.50–62.47) | | 6 (7.14%) | 0.35 (0.00–26.33) | | 3 (5.45%) | 0.33 (0.01–10.87) | |
| Extremity | 14 (4.5%) | 2 (1.98%) | 3.71 (0.10–143.00) | | 10 (14.08%) | 8.45 (0.53–135.81) | | 1 (1.19%) | 0.36 (0.00–285105.30) | | 1 (1.82%) | 0.35 (0.00–49.28) | |
| Intra-abdominal | 62 (19.94%) | 24 (23.76%) | 0.82 (0.18–3.78) | | 14 (19.72%) | 6.56 (0.38–114.47) | | 12 (14.29%) | 4.29 (0.31–58.62) | | 12 (21.82%) | 1.71 (0.25–11.56) | |

**Table 2 (continued) | Univariate analysis for PFS in SYSUCC cohort, FAHSYSU cohort, CHCAMS cohort 1 and CHCAMS cohort 2 of primary non-CNS SFTs with negative tumor margins**

| Clinicopathologic feature | Total number (%) | SYSUCC cohort (n = 101) | | | FAHSYSU cohort (n = 71) | | | CHCAMS cohort 1 (n = 84) | | | CHCAMS cohort2 (n = 55) | | |
|---|---|---|---|---|---|---|---|---|---|---|---|---|---|
| | | Number (%) | Univariate analysis HR (95% CI) | P value | Number (%) | Univariate analysis HR (95%CI) | P value | Number (%) | Univariate analysis HR (95%CI) | P value | Number (%) | Univariate analysis HR (95% CI) | P value |
| Radiotherapy | | | | 0.228 | | | 0.001 | | | 5.81E-08 | | | 0.342 |
| Yes | 8 (2.57%) | 3 (2.97%) | 3.24 (0.10–105.24) | | 2 (2.82%) | 14.61 (0.01–18112.95) | | 1 (1.19%) | 36.89 (0.00–2052714.39) | | 2 (3.64%) | 2.68 (0.11–62.71) | |
| No | 303 (97.43%) | 98 (97.03%) | 1.00 | | 69 (97.18%) | 1.00 | | 83 (98.81%) | 1 | | 53 (96.36%) | 1.00 | |
| Chemotherapy | | | | 0.234 | | | 0.341 | | | 5.81E-08 | | | 3.28E-11 |
| Yes | 10 (3.22%) | 4 (3.96%) | 3.22 (0.10–103.00) | | 4 (5.63%) | 2.62 (0.11–60.25) | | 1 (1.19%) | 36.89 (0.00–2052714.39) | | 1 (1.82%) | 52.33 (0.00–27409247.95) | |
| No | 301 (96.78%) | 97 (96.04%) | 1.00 | | 67 (94.37%) | 1.00 | | 83 (98.81%) | 1 | | 54 (98.18%) | 1.00 | |
| Follow-up time (months) | 41.10 (0.00–171.30) | 50.40 (1.50–171.30) | | | 32.30 (0.00–121.20) | | | 36.00 (0.00–73.90) | | | 72.60 (0.30–121.10) | | |

*HR* Hazard Ratio, *CI* Confidence Interval, *D* Diameter, *HPF* High Power Field, *WHO* World Health Organization, *NOS* Not Otherwise Specified.
P values were calculated using two-sided log-rank test.

Tables 3, 9, 10, Supplementary Fig. 6a–c), while the integrated model didn't work in the cohort of relapsed non-CNS SFTs with NTM (*n* = 31) (Supplementary Tables 4, 11, Supplementary Fig. 6d–e), and it may be attribute to the complexity of relapsed SFTs cases. Collectively, our integrated risk model could stratify the primary SFTs with NTM originated from both non-CNS and CNS into low-risk group, intermediate-risk group, and high-risk group efficiently.

### Comparison of integrated model to WHO classification and published models

The C-index of integrated model, WHO classification, mDemicco model and G-score in discovery SYSUCC cohort (*n* = 101) were 0.911 (95% CI, 0.846–0.976), 0.787 (95% CI, 0.660–0.914), 0.857 (95% CI, 0.763–0.951), 0.855 (95% CI, 0.749–0.961) respectively (Table 4). Moreover, the AUC of integrated model in SYSUCC cohort was 0.921 (95% CI, 0.868–0.974), which was greater than that of WHO classification, mDemicco model, and G-score (0.921 vs 0.790/0.832/0.865) (Fig. 5a). In addition, both C-index and AUC of the integrated model in the FAHSYSU cohort (Table 4, Fig. 5b), CHCAMS cohort 1 (Table 4, Fig. 5c), and CHCAMS cohort 2 (Table 4, Fig. 5d) were also greater than that of the WHO classification, mDemicco model, and G-score. Collectively, both the C-index and AUC of the integrated model were greater than those of the WHO classification, mDemicco model and G-score in the discovery cohort of SYSUCC and in three validation cohorts of FAHSYSU and CHCAMSs.

In addition to PFS, metastasis free survival (MFS) and recurrence free interval (RFI) were also used for comparison, and similar results were obtained (Supplementary Table 12, Supplementary Fig. 7a–c). The integrated model stratified all primary non-CNS SFTs with NTM in four cohorts (*n* = 311) as low-risk, intermediate-risk, and high-risk groups at 50.80% (158/311), 35.69% (111/311), and 13.50% (42/311), respectively, and the corresponding 5 years PFS was 100%, 73%, 21% (*p* = 2.00E−16, Supplementary Table 13, Supplementary Fig. 7d). However, the WHO classification, mDemicco model and G-score model stratified all SFTs in four cohorts (*n* = 311) as low-risk (or locally aggressive, benign), intermediate-risk (or rarely metastasizing, nos), and high-risk groups (or malignant) at 78.14%/4.82%/17.04%, 73.63%/19.94%/6.43%, and 37.62%/45.66%/16.72%, respectively (Supplementary Table 13, Supplementary Fig. 7e–g). The correlation between the integrated model, WHO classification, mDemicco model and G-score was poor (Supplementary Fig. 7h–j), which was consistent with previous reports[14–16].

Since our integrated model was successfully validated in primary CNS SFTs with NTM (*n* = 36) (Supplementary Tables 3, 9, Supplementary Fig. 6a–c), we compared the prediction efficacy between integrated model with WHO grade based on C-index and AUC. We found that the C-index of integrated model was greater than that of WHO grade (0.901 vs 0.802, Supplementary Table 10), while the AUC of integrated model was also greater than that of WHO grade (0.867 vs 0.756, Supplementary Fig. 6a). Taken together, our integrated model showed better accuracy and improved efficacy in identifying patients with primary non-CNS and CNS SFT with NTM at risk for progression, compared to the current WHO classification / grade, mDemicco model, and G-score.

## Discussion

Managing patients with SFT presents challenges due to the lack of an accurate risk-predicting model and the absence of clearly effective systemic or targeted therapies that have been conclusively proven[16]. Published data on the response of SFTs to conventional chemotherapy are limited, and findings from small case series, retrospective studies, and predictive preclinical models have yielded conflicting results. While some small-scale clinical trials on targeted therapy, such as anti-angiogenic agents, have been reported, further exploration is still necessary. As a result, there are currently no standardized guidelines for early accurate prediction and systemic treatment of SFTs, and

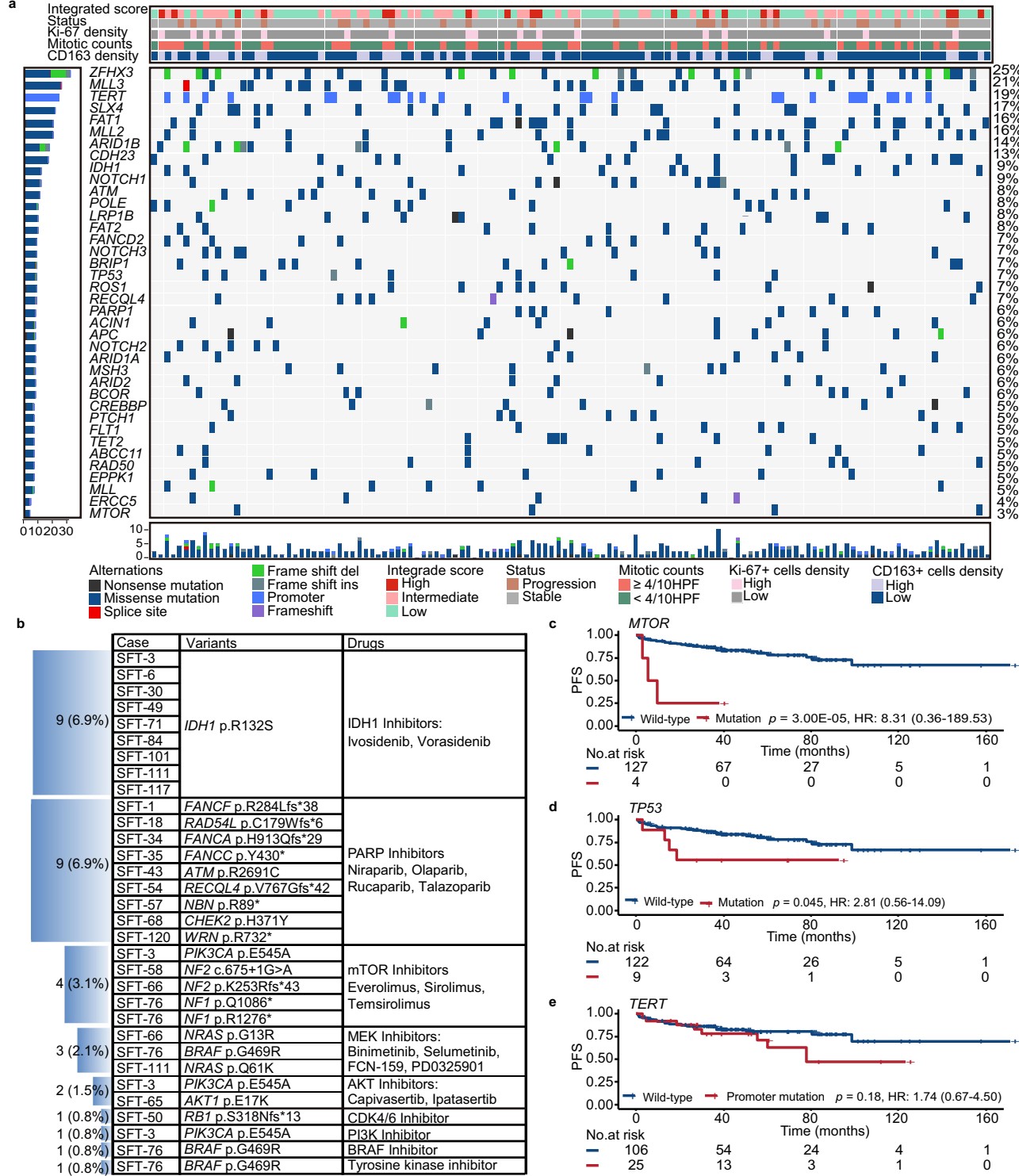

**Fig. 2 | Genomic landscape analysis identified clinical actionable targets for SFTs. a** High frequency of altered genes across 131 cases of SFTs in the discovery cohort; **b** Clinical actionable targets were identified in SFTs, notably *IDH1* p.R132S variant was found in 6.9% of SFTs (9/131); Kaplan–Meier plots showing PFS for patients who had *MTOR* mutation and wild type (**c**), *TP53* mutation and wild type (**d**), *TERT* promoter mutation and wild type (**e**). *p* values were calculated using two-sided log-rank test. SFTs Solitary Fibrous Tumors, HR Hazard Ratio. Source data are provided as a Source Data file.

further research into therapeutic targets, risk-predicting models, and additional treatment options is highly desired.

In the present study, we identified *IDH1* p.R132S in 6.9% (9/131) of SFTs. Interestingly, *IDH1* p.R132S mutation was highly enriched in the WHO classification of malignant, high nuclear pleomorphism, and high cellularity cases of SFT. Currently, there are few studies on the genomic landscape of SFT[1,2,24–28], and *IDH1* mutations have not yet been

reported yet probably due to limited sample size, relatively low frequency of *IDH1* mutations, and technical issues (the NGS panel in some studies did not include *IDH1* gene). *IDH1* and *IDH2* are frequently mutated in multiple types of human cancers, including glioma, acute myeloid leukaemia, cholangiocarcinoma, chondrosarcoma, and thyroid carcinoma[29,30]. To date, dozens of small molecules of IDH1/IDH2 inhibitors are under investigation, of which ivosidenib (AG-120) has

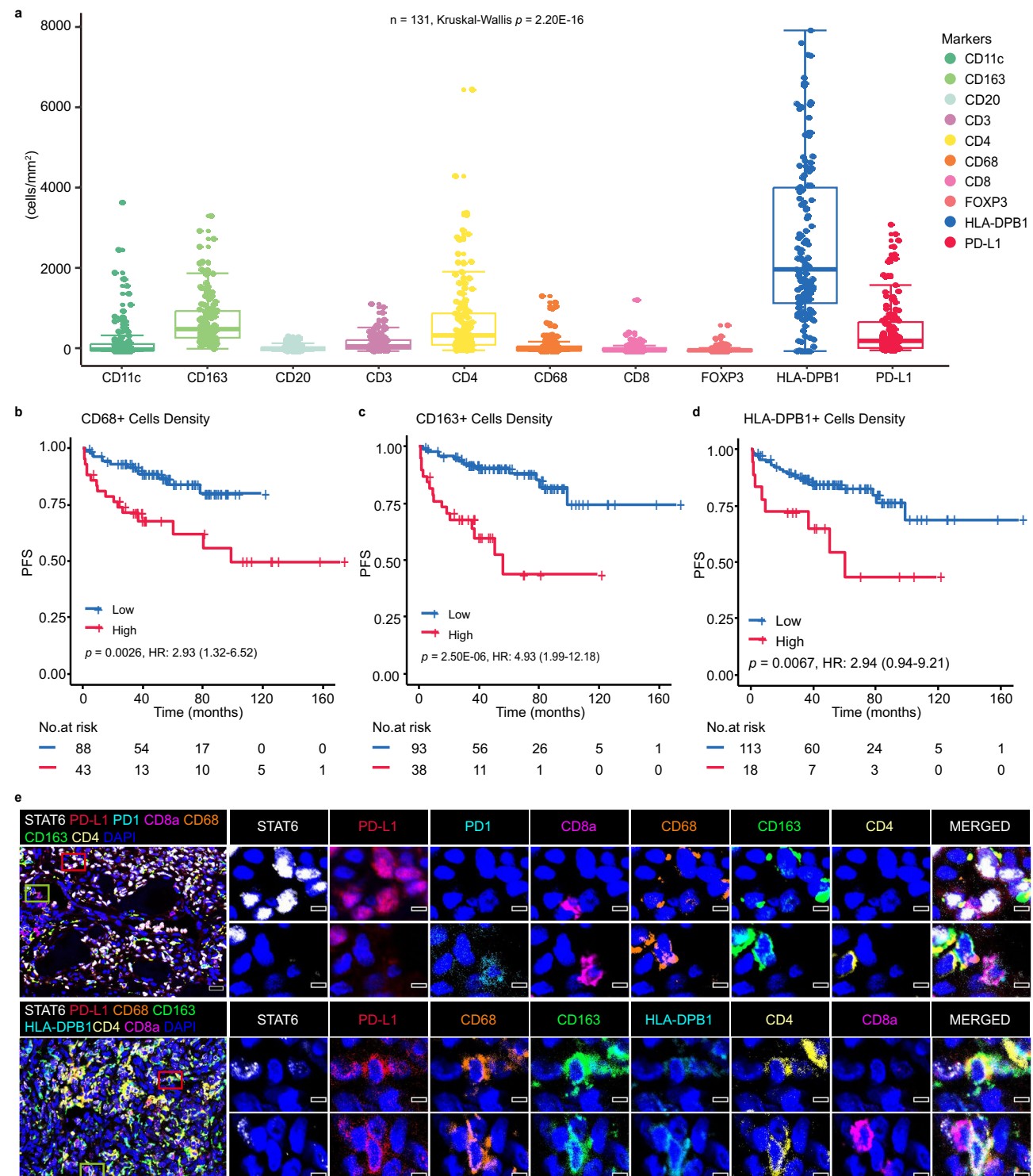

**Fig. 3 | High density of macrophages and high PD-L1 expression were found in SFTs. a** Boxplot indicating that the tumor infiltrating immune cells in SFTs (SYSUCC cohort, *n* =131) are predominantly macrophages. Box center lines, bounds of the box, and whiskers indicate medians, first and third quartiles, and minimum and maximum values within 1.5×IQR (interquartile range) of the box limits, respectively. *p* values were calculated using Kruskal−Wallis test. **b**−**d** High density of CD68+ macrophages, CD163+ macrophages, and HLA-DPB1+ cells infiltrated in SFTs tissues was associated with shorter PFS of the patients. *p* values were calculated using two-sided log-rank test. **e** PD-L1 was highly expressed in both tumor cells and CD68+/HLA-DPB1+/CD163+ macrophages in SFTs (scale bars: 100 μm/5 μm). Multiplex immunofluorescence was performed on three cases of SFT tissues and the representative images are presented here. HR Hazard Ratio, PFS Progression Free Survival. Source data are provided as a Source Data file.

been approved for the treatment of *IDH1* mutated cholangiocarcinoma and acute myeloid leukaemia[31–33]. Therefore, targeting *IDH1* p.R132S could play a crucial role and offer an additional treatment option for high-grade SFTs.

In this study, we found that the tumor-infiltrating immune cells in SFTs were predominantly macrophages, and high infiltration of macrophages was significantly associated with short PFS of the patients. Furthermore, we observed that 24.4% (32/131) of SFTs

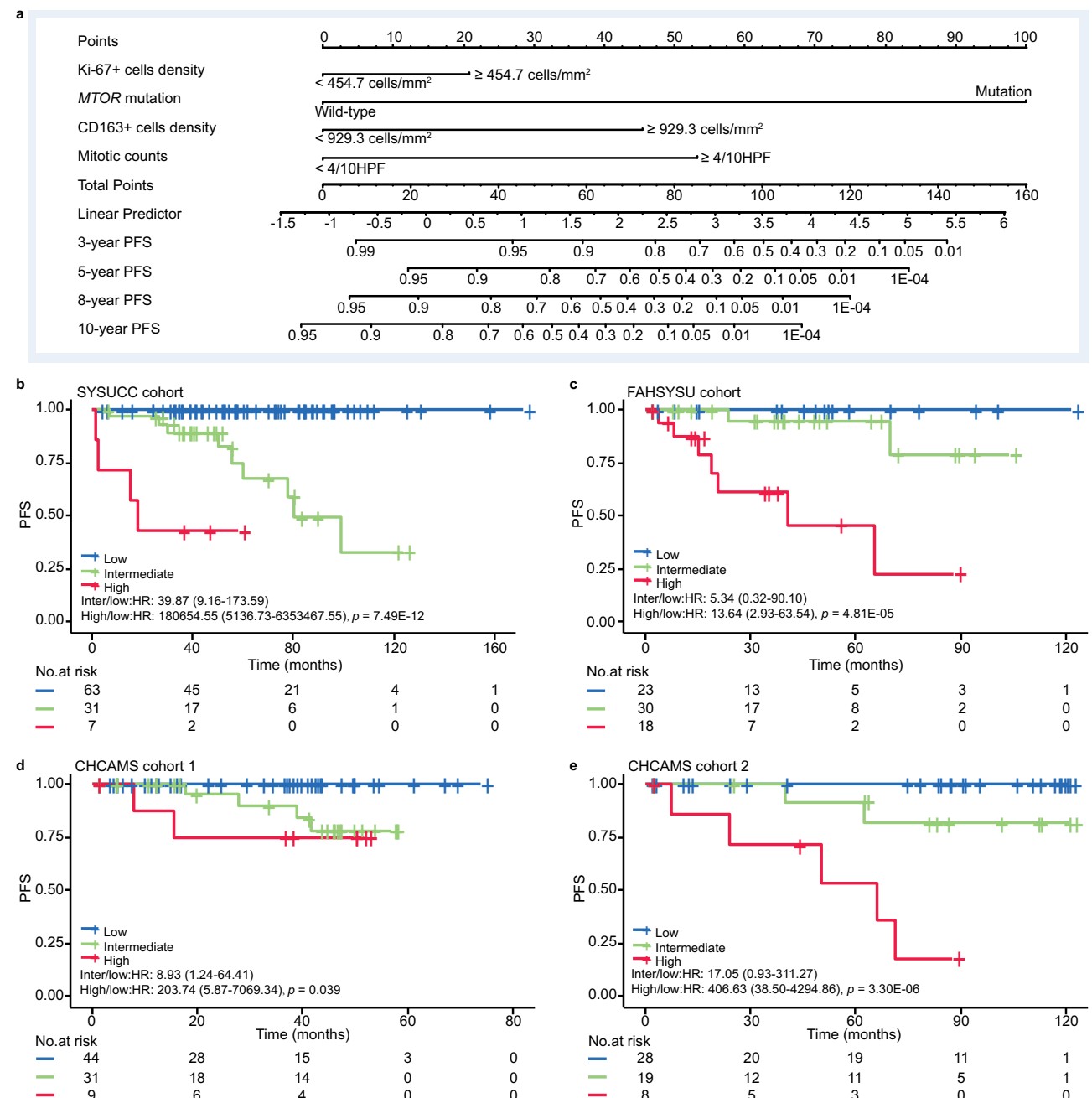

**Fig. 4 | Development of an integrated risk model in the discovery cohort and validation in three independent cohorts. a** The nomogram of the integrated risk model incorporates four variables: mitotic counts, the density of Ki-67+ and CD163+ cells, and *MTOR* mutation. It predicts the 3-, 5-, 8-, and 10-year progression risk for each SFT patient. The importance of each variable is ranked based on the standard deviation along the nomogram scales. To use the nomogram, specific points corresponding to individual patients are located on each variable axis. The sum of these points is then located on the total points axis, and a line is drawn downward to the survival axes to determine the probability of 3-, 5-, 8-, and 10-year PFS. Kaplan–Meier plots showing PFS for SFT patients stratified by integrated risk model in: **b** SYSUCC cohort. **c** FAHSYSU cohort. **d** CHCAMS cohort 1. **e** CHCAMS cohort 2. *p* values were calculated using two-sided log-rank test. HR Hazard Ratio, PFS Progression Free Survival. Source data are provided as a Source Data file.

exhibited high PD-L1 expression in either tumor cells or in immune cells. Notably, increased density of CD20+ B cells was enriched in SFTs with tumors expressing PD-L1+, and associated with long PFS of the patients. In-depth characterization of the spatial relationships between CD8+ T cells, CD20+ B cells, macrophages and other immune cells, as well as further classification of certain immune cells, would be of great interest for future studies in SFTs. In this regard, the application of highly multiplexed tissue imaging technologies, such as imaging mass cytometry (IMC)[34], multiplexed ion beam imaging (MIBI)[35], co-detection by indexing (CODEX)[36], and

t-CyCIF[37,38], could be valuable in measuring the distance between different cell types within the tumor microenvironment. However, current implementation of these technologies on a large-scale sample for outcome analysis is challenging due to the slow scanning speed, high cost, and technical complexity involved. Nevertheless, these advanced imaging approaches hold great promise for advancing our understanding of tumor immune microenvironment in SFTs in the future.

The PD-1/PD-L1 axis inhibition alone is not enough for SFTs treatment, and double inhibition of angiogenesis and PD-1/PD-L1 axis

was demonstrated as an effective treatment strategy, supporting combination strategies promoting inflamed microenvironment resulted in a higher efficacy in SFTs[39]. In addition, the combination of colony stimulating factor 1 receptor (CSF1R) inhibitor with anti-PD-L1/PD-1 axis blockage has been reported to be highly active in both melanoma and hepatocellular carcinoma mouse models by elimination of tumor-associated macrophages (TAMs)[40,41]. Notably, the activity of pazopanib was demonstrated in SFTs by two prospective phase II studies with overall response rate of about 50% and pazopanib was administered from first line therapy recommended by ESMO-EURACAN-GENTURIS clinical practice guideline[42–44]. In present study, interestingly, we found that two SFT cases showed both *IDH1* p.R132S mutation and high PD-L1 expression. Hence, the different combination of PD-1/PD-L1 axis inhibition with anti-angiogenesis, anti-macrophages, or anti-IDH1 treatment based on biomarkers screening may be a treatment strategy in future trials for SFTs.

Several risk stratification models, which usually include only clinical and histopathological factors, have been developed to identify the high risk of non-CNS SFTs. Demicco et al. developed a widely used scoring system based on age, tumor size, mitotic count and necrosis to predict metastasis[10,11]. Georgiesh and colleagues developed a G-score risk model that included mitotic count, necrosis, and sex to account for both early and late recurrences and was shown to predict recurrence risk[12,13]. Unfortunately, those risk models do not accurately predict clinical evolution of SFTs[14–16,45,46]. In our study, we sought to address this gap by developing an integrated risk model comprising four variables: mitotic count, the density of Ki-67+ and CD163+ cells, and *MTOR* mutation, and compared our integrated model with the WHO classification/WHO grade and two published models, the mDemicco model and G-score. Our results indicated that both the C-index and AUC of the integrated model were greater than those of the WHO classification/WHO grade, mDemicco model and G-score in the discovery cohort of SYSUCC, and in the three validation cohorts. Consequently, our integrated model represents a significant improvement in accurately identifying patients with primary non-CNS and CNS SFT with NTM who are at risk of disease progression, compared to the WHO classification, mDemicco model, and G-score.

One notable strength of our study is the comprehensive assessment of both mutation status and other gene variants (amplification, deletion, fusion) in 1021 cancer-related genes, conducted in a relatively large cohort of SFTs ($n = 131$). This approach allowed us to explore the genomic landscape and identify *IDH1*-mutated SFTs. Another key strength lies in the utilization of relatively large cohorts for both model development and validation. Our integrated risk model was initially generated in the discovery cohort, which consisted of 101 cases. Subsequently, we validated the model in three independent cohorts with varying follow-up durations, encompassing a total of 210 primary non-CNS SFTs with NTM, as well as in a separate cohort of 36 primary CNS SFTs with NTM. Notably, our model is unique in that it is applicable to both non-CNS and CNS SFTs with NTM, whereas existing published models are solely designed for non-CNS SFTs. However, the study also has some limitations. First, we did not analyze matched normal DNA, which could have enhanced the detection of mutations. In addition, the absence of RNA-based NGS may result in the loss of information on some fusion genes, such as the *NAB2-STAT6* fusion gene. Furthermore, our patient cohorts included most SFTs that had been resected within the past several years. Therefore, long-term follow-up still needed to assess clinical outcomes is not yet available. Nonetheless, the robust validation of our integrated risk model across multiple independent cohorts underscores its potential clinical utility and reliability for predicting tumor progression in both non-CNS and CNS SFTs with NTM.

Our integrated risk model has demonstrated superior accuracy in identifying high-risk primary non-CNS and CNS SFTs with NTM, and showed better performance compared to established models such as the WHO classification, mDemicco model, and G-score. In addition, the applicability of our integrated model to both CNS and non-CNS SFTs broadens its clinical application potential. However, the practical implications of our integrated model must be thoroughly evaluated in the context of clinical care and therapeutic decision-making. The potential clinical benefits and justifications for the additional investigations required were discussed as below: (1) Enhanced therapeutic decision: With targeted gene NGS testing for mutation information becoming routine in both academic institutions and commercial entities, the incorporation of *MTOR* mutation status into our model is increasingly feasible. Moreover, the identification of potential therapeutic targets by NGS, such as the *IDH1* p.R132S mutation, and the

**Table 3 | Integrated risk stratification model for solitary fibrous tumors**

| Risk factor | Score (Linear Predictor) | Points |
|---|---|---|
| **Mitotic counts** | | |
| <4/10HPF | 0 | 0 |
| ≥4/10HPF | 2.42 | 53 |
| ***MTOR* mutation** | | |
| Wild-type | 0 | 0 |
| Mutation | 4.56 | 100 |
| **Ki-67+ cells density (cells/mm²)** | | |
| <454.7 (190 cells/10HPF)[a] | 0 | 0 |
| ≥454.7 (190 cells/10HPF) | 0.949 | 21 |
| **CD163+ cells density (cells/mm²)** | | |
| <929.3 (387cells/10HPF)[b] | 0 | 0 |
| ≥929.3 (387cells/10HPF) | 2.07 | 45 |
| **Risk stratification** | **Total score** | **Total points** |
| Low | 0 | 0 |
| Intermediate | 0–3.37 | 0–74 |
| High | >3.37 | 74 |

*HPF* High Power Field.

[a,b]The common field number (F.N.) for a microscope is 22 mm, the area under the visual field of 40× was 0.24 mm², 454.7cells/mm² equals 190 cells/10HPF, 929.3 cells/mm² equals 387cells/10HPF.

**Table 4 | C-index of integrated risk model, WHO classification, mDemicco model, and G-score in discovery SYSUCC cohort and three validation cohorts**

| Classification | SYSUCC cohort ($n = 101$) | | FAHSYSU cohort ($n = 71$) | | CHCAMS cohort 1 ($n = 84$) | | CHCAMS cohort 2 ($n = 55$) | |
|---|---|---|---|---|---|---|---|---|
| | C-index (95%CI) | *P* value | C-index (95%CI) | *P* value | C-index (95%CI) | *P* value | C-index (95%CI) | *P* value |
| Integrated model | 0.911 (0.846–0.976) | 1.16E–35 | 0.825 (0.727–0.923) | 9.74E–11 | 0.890 (0.786–0.994) | 1.78E–13 | 0.903 (0.803–1.002) | 3.40E–15 |
| WHO classification | 0.787 (0.660–0.914) | 9.91E–06 | 0.680 (0.501–0.859) | 4.90E–02 | 0.838 (0.672–1.004) | 1.00E–04 | 0.785 (0.615–0.955) | 1.00E–03 |
| mDemicco model | 0.857 (0.763–0.951) | 8.24E–14 | 0.760 (0.601–0.919) | 1.00E–03 | 0.884 (0.807–0.961) | 1.99E–22 | 0.814 (0.674–0.954) | 1.17E–05 |
| G-score | 0.855 (0.749-0.961) | 5.01E–11 | 0.785 (0.637–0.933) | 2.00E–04 | 0.822 (0.655–0.989) | 1.00E–04 | 0.850 (0.747–0.953) | 2.93E–11 |

*CI* Confidence Interval, *WHO* World Health Organization.

*P* values were calculated using two-sided *z* test.

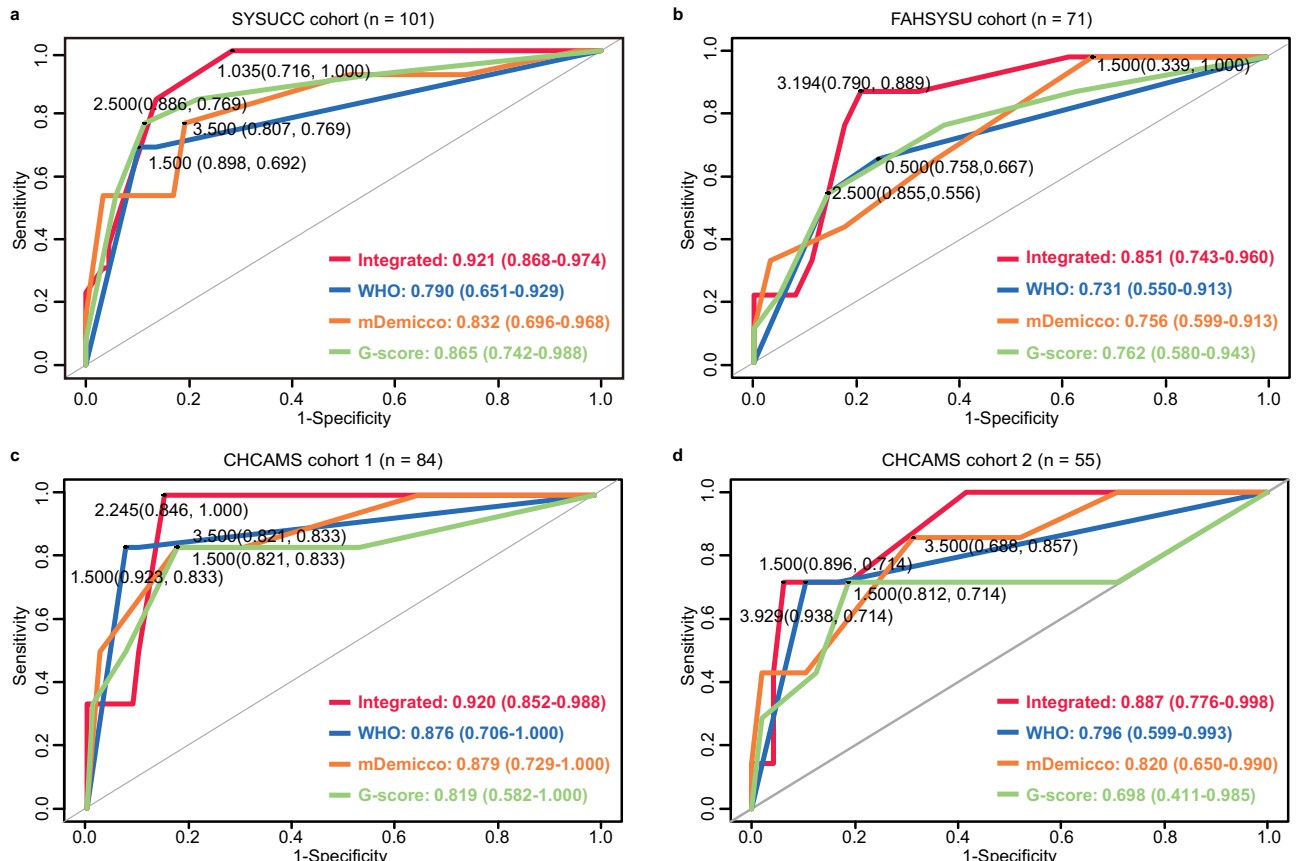

**Fig. 5 | Comparison of the integrated risk model to WHO classification and published models.** ROC curves of integrated risk model, WHO classification, mDemicco model, and G-score in (**a**) discovery SYSUCC cohort, (**b**) FAHSYSU cohort, (**c**) CHCAMS cohort 1, (**d**) CHCAMS cohort 2. WHO World Health Organization. Source data are provided as a Source Data file.

detection of CD163+ macrophages and PD-L1 expression by IHC, could stimulate further research into treatments for SFTs, such as combinational immune therapy and targeted therapy. (2) Clinical risk stratification: Our model provides a more accurate assessment of a patient's risk of tumor progression, enabling improved risk stratification. This information can guide clinicians in deciding the frequency of follow-up assessments and the intensity of surveillance, and optimize healthcare resource allocation. Our model could identify high-risk patients at an earlier stage, allowing for more aggressive treatment strategies that could lead to better outcomes. By accurately identifying low-risk patients, our model helps avoid unnecessary over-treatment, reducing both patient risk and healthcare costs.

In conclusion, our study has identified the mutation of *IDH1* p.R132S and PD-L1 expression as potential therapeutic targets for SFTs, offering the possibility of more precise treatment options through combination targeted therapy or immunotherapy based on biomarkers screening. In addition, we have proposed an integrated risk stratification model for primary non-CNS and CNS SFTs with NTM, allowing for the prediction of tumor progression. This integrated model holds significant promise and practicality for clinicians, representing a significant advancement in the treatment of patients with SFTs.

## Methods
### Ethics approval
This study was approved by the regional ethics committees at all participating institutions (Ethics Committee of Sun Yat-sen University Cancer Center: SYSUCC, B2021-421-01; Ethics Committee of The First Affiliated Hospital, Sun Yat-sen University: FAHSYSU, [2022]409; Ethics

Committee of National Cancer Center/Cancer Hospital, Chinese Academy of Medical Sciences and Peking Union Medical College: CHCAMS, 22/024-3225). The requirement for informed consent was waived, and no compensation was provided to the participants in this study.

### Patients and tumor tissues
We collected formalin-fixed paraffin-embedded (FFPE) tumor tissue specimens from 131 histologically proven SFT cases at the Sun Yat-sen University Cancer Center (SYSUCC) between 2008 and 2020, forming the discovery cohort (SYSUCC cohort). In addition, we recruited 277 FFPE tumor tissue specimens from two other hospitals to establish three independent validation cohorts. These validation cohorts included 115 cases from the First Affiliated Hospital of Sun Yat-sen University between 2013 and 2021 (FAHSYSU cohort), 101 cases from the Cancer Hospital Chinese Academy of Medical Sciences between 2017 and 2021 (CHCAMS cohort 1), and 61 cases from CHCAMS between 2013 and 2016 (CHCAMS cohort 2) (Table 1). The inclusion criteria were as follows: (i) All tumors were histologically verified as SFT. (ii) None of the tumor samples received chemotherapy, radiotherapy, or targeted therapy prior to surgery. (iii) All the patients had complete clinicopathological data.

Clinical information, including patient sex (sex was considered in the study design and sex of participants was determined by self-report and biology characteristics), age at diagnosis, specimen type, tumor size, tumor site, tumor margin, adjuvant treatment and patient outcome was retrieved from the patient medical records. Specimen type: primary tumor, recurrent or metastatic tumor; patient age at diagnosis: ≥55 years or <55 years[10]; tumor size was taken as the largest gross tumor measurement on the resected specimen, ≥5 cm, 5–10 cm, 10–15 cm, or

≥15 cm[10]; tumor site was categorized as intra thoracic, head and neck, trunk, extremity, intra-abdominal and central nervous system (CNS); tumor margin was classified as R0 (microscopic complete), R1 (microscopic incomplete) and R2 (macroscopic incomplete) according to the Union for International Cancer Control (UICC) criteria[7]; the primary outcome assessed was PFS, defined as the time from surgery to disease progression (recurrence or metastasis) or death. The clinicopathological information of the patients was listed in Table 1.

## Histopathological evaluation

All hematoxylin and eosin (H&E) slides of individual specimens were reviewed and re-classified by experienced soft tissue pathologists (YF, YY, CH) using the fifth edition of WHO classification for non-CNS SFTs[5] and CNS SFTs[6] respectively. The following histopathological data from each case were also retrieved: mitotic counts were calculated per 10 High Power Fields (HPF, ×400) in the most mitotically active area of the tumor, <4/10HPF or ≥4/10HPF; tumor necrosis was scored as positive when involving 10% or more of the tumor, except hemorrhage or hyalinization; nuclear pleomorphism was scored as low (cells monomorphic, with uniform nuclear features), moderate (increased nuclear pleomorphism, more prominent nucleoli and rare multinucleated cells), or high (hyperchromatic nuclei present with foci of marked pleomorphism and bizarre cells); cellularity was scored as low (tumor predominately composed of sclerotic collagen bands with scattered, compressed spindle cells), moderate (many areas of increased cellularity with cells adjacent to one another) and high (hypercellular tumor, with areas of nuclear overlap). Histopathological features were evaluated as described previously[10,11].

## Immunohistochemistry and evaluation

The following primary antibodies were used in this study: STAT6 (Ready to use, EP325, Cat# RMA-0845, MXB Biotechnologies), Ki-67 (Ready to use, MXR002, Cat# RMA-0731, MXB Biotechnologies), CD68 (Ready to use, KP1, Cat# Kit-0026, MXB Biotechnologies), CD163 (Ready to use, 10D6, Cat# MAB-0206, MXB Biotechnologies), HLA-DPB1 (1:3000 dilution, EPR11226, Cat# ab157210, Abcam), PD-L1 (1:500 dilution, E1L3N, Cat# 13684S, Cell signaling technology), CD3 (Ready to use, SP7, Cat# Kit-0003, MXB Biotechnologies), CD4 (Ready to use, SP35, Cat# RMA-0620, MXB Biotechnologies), CD8 (Ready to use, SP16, Cat# RMA-0514, MXB Biotechnologies), FOXP3 (1:400 dilution, 206D, Cat# 320102, Biolegend), CD11c (1:300 dilution, EP1347Y, Cat# ab52632, Abcam), and CD20 (Ready to use, L26, Kit-0001, MXB Biotechnologies). MaxVision TM HRP-Polymer anti-Mouse/Rabbit IHC Kit, Secondary Antibody (Ready to use, Cat# KIT-5020, MXB Biotechnologies) was used as secondary Antibody. All antibodies were validated before experiments. Detailed information on these antibodies was provided in Supplementary Table 14. We used 3,3′-diaminobenzidine (DAB) for visualization, and the sections were counterstained with hematoxylin. The slides were scanned using a digital pathology scanner (Axio Scan.Z1, Germany) to acquire digital images that were quantitatively scored using the HALO 2.3 digital pathology system (Indica Labs).

A study pathologist (RZ) evaluated STAT6 staining by light microscopy using a method described previously[18]. To characterize the immune infiltrate and proliferative cells in SFTs, we counted Ki-67, CD68, CD163, HLA-DPB1, CD3, CD4, CD8, FOXP3, CD11c, CD20 positive cells using the HALO 2.3 digital pathology system (Indica Labs). The average density of positive cells (cells/mm²) in the tumor area of each case was calculated. The optimal cut-off value was defined using $X$-tile software, a tool for biomarker assessment and outcome-based cut-point optimization[47]. Cases with an average density of positive cells values exceeding the cut-off value were categorized as having a high infiltrate of a particular marker, while cases with an average density of positive cells values below the cut-off value were classified as having a low infiltrate of that marker.

To evaluate PD-L1 expression in SFTs, we combined digital quantification using HALO and visual review by a study pathologist (RZ). Initially, the HALO 2.3 digital pathology system (Indica Labs) was utilized to identify PD-L1 positive cells, and subsequently, the study pathologist verified the cell types of these positive cells, classifying them as either tumor cells or immune cells. For cases with PD-L1 expressing tumor cells, those with a proportion of PD-L1 positive cells (including tumor cells, macrophages, or lymphocytes) ≥20% of the total cell count were considered to have high PD-L1 expression in tumor cells. For cases with PD-L1 expressing immune cells, those with a proportion of positive immune cells (PD-L1 stained immune cells) ≥25% of the total immune cell count were categorized as having high PD-L1 expression in immune cells.

## Multiplex immunofluorescence

Multiplex immunofluorescence staining, as described previously was carried out to detect the subtype of immune cells expressing PD-L1. Slides underwent multiple cycles of antibody incubation, imaging, and fluorophore inactivation[38]. Briefly, antibodies were incubated overnight at 4 °C in the dark, and hoechst 33342 was used for DNA staining. Glass coverslips were wet-mounted using 100 μL of 70% glycerol in 1× PBS. Images were acquired using a confocal microscope (LSM 980, Carl ZEISS, Germany) with a 20 × / 0.80 NA objective. The fluorophores were then inactivated by incubating slides in a solution of 4.5% $H_2O_2$ and 24 mmol/L NaOH in PBS and placing them under an LED light source for 2 h. The following fluorescently labeled antibodies were used in this study: CD68 (1:100 dilution, D4B9C, Cat# 79594 S, Cell Signaling Technology), CD163 (1:100 dilution, EPR14643-36, Cat# ab218294, Abcam), CD206 (1:100 dilution, D-1, Cat# sc-376108, Santa Cruz Biotechnology), STAT6 (1:100 dilution, YE361, Cat# ab207014, Abcam), PD-L1 (1:100 dilution, SP142, Cat# ab267563, Abcam), PD1 (1:100 dilution, EPR4877(2), Cat# ab201825, Abcam), CD4 (1:100 dilution, N1UG0, Cat# 41-2444-80, eBioscience) and CD8a (1:100 dilution, AMC908, Cat#50-0008-80, eBioscience). Chicken anti-Goat IgG (H + L) Cross-Adsorbed Secondary Antibody, Alexa Fluor 488 (1:500 dilution, Cat# A-21467, ThermoFisher), Goat anti-Rabbit IgG (H + L) Cross-Adsorbed Secondary Antibody, Alexa Fluor 555 (1:500 dilution, Cat# A-21428, ThermoFisher), Chicken anti-Mouse IgG (H + L) Cross-Adsorbed Secondary Antibody, Alexa Fluor 647 (1:500 dilution, Cat# A-21463, ThermoFisher) were used to label primary antibody. All antibodies were validated before experiments. Detailed information on these antibodies was provided in Supplementary Table 14. Hoechst 33342 (Cat# 4082S, Cell Signaling Technology) for DNA staining. Images were acquired using a confocal microscope (LSM 980, Carl ZEISS) with a 20 ×0.80 NA objective.

## DNA isolation, DNA sequencing, and RT-PCR

FFPE tumor tissue specimens from each patient were cut into 4–5 μm thick sections. Corresponding H&E slides were used for enrichment of tumor regions containing >80% of tumor cells. Genomic DNA was extracted using a DNA FFPE Tissue kit (Cat# DP214-03, TIANGEN, China).

Targeted next-generation sequencing (NGS) was performed at clinical laboratory in the Department of Molecular Diagnostics at Sun Yat-sen University Cancer Center using a pan-cancer 1021-gene panel (Geneplus-Beijing institute, China) for the simultaneous detection of 1021 cancer related genes[22] in the discovery cohort ($n = 131$). For library construction, ~0.5 μg of DNA fragments were mixed with Illumina-indexed adapters (Illumina, San Diego, CA, USA) using the KAPA Library Preparation Kit (Kapa Biosystems, Wilmington, MA, USA). A hybrid captured-based NGS assay covering ~1.1 megabases (Mb) of the genomic sequences of 1021 cancer-related genes (Cat# DC204802, GenePlus-Beijing, China) was used for the sequencing, which was performed using a GenePlus 2000 sequencing system (Geneplus-Beijing institute, China) with 2 × 100 bp paired-end reads. BWA18 (version 0.7.12-r1039) was used to align clean reads to the reference

human genome (hg38). Single-nucleotide variants (SNVs) and small insertions and deletions (indels) were identified using MuTect19 (version 1.1.4). Mutations were annotated to genes using the ANNOVAR20 software. CONTRA21 was used to detect copy number variations (CNVs). The clinical significance of sequence variants was categorized following the Standards and Guidelines for the Interpretation and Reporting of Sequence Variants in Cancer[48]. Potentially damaging mutations of sequence variants were predicted by both PolyPhen2/SIFT (ensdb v73) and Cosmic (V80)[49,50]. Effective mutations of specific genes were defined as damaging mutations predicted by either PolyPhen2/SIFT (ensdb v73) or Cosmic (V80) in this study.

Sanger sequencing was performed to detect *MTOR* gene mutations in SFT specimens from three independent validation cohorts ($n = 277$) and *IDH1* p.R132S mutations in seven SFT specimens from the SYSUCC cohort. In general, gDNA was used to amplify two fragments of *MTOR* gene or one fragment of *IDH1* gene by standard PCR (Polymerase Chain Reaction), using the specific primers listed in Supplementary Table 15. The PCR products were sequenced using ABI 3500XL Genetic Analyzer (Applied Biosystems, USA).

The detection of *NAB2-STAT6* fusion transcript by RT–PCR was performed in 11 cases with STAT6 IHC negative in SYSUCC cohort. In brief, five 10-μm-thick tissue scrolls were cut from each representative FFPE block for RNA extraction by RNeasy FFPE Kit (Cat# No.73504, Qiagen, Germany). The yield and quality of mRNA were determined by a nanodrop UV spectrophotometer. FastKing gDNA Dispelling RT SuperMix (Cat# KR118-02, TIANGEN, China) was used to synthesize the cDNA, and PCR was performed by Platinum Taq DNA polymerase (Cat# RR390A, TaKaRa, Japan) with the primers targeting different types of *NAB2-STAT6* fusion listed in Supplementary Table 16[51]. The PCR products were sequenced by ABI 3500XL Genetic Analyzer (Applied Biosystems, USA).

### Risk modeling and comparison

The variables used for analysis initially included clinical and histopathological factors (including, mitotic count, necrosis, age, sex, tumor size, tumor site, cellularity and nuclear pleomorphism), immunohistochemical factors (including the density of cells of Ki-67+, CD68+, CD163+, HLA-DPB1+, PD-L1+, CD3+, CD4+, CD8+, FOXP3+, CD11c+, CD20+, respectively) and molecular factors (including *MTOR* mutation, *NOTCH* pathway mutation, *ERCC5* mutation, *TP53* mutation, *TERT* promoter region mutation and *IDH1* mutation). To select the most relevant variables, we employed LASSO regression and random survival forest (RandomForestSRC R package v3.2.0) methods. Variables predicted by both methods were included in the model based on their importance ranking, and this process continued until the addition of variables no longer improved the model's performance.

For non-CNS SFTs with NTM, we compared the integrated model with WHO classification, and two previously published models, including the modified Demicco model (mDemicco model, variables including mitotic count, patient age, tumor size and necrosis)[11], and G-score (variables including mitotic count, necrosis and patient sex)[13], which have been developed for non-CNS SFTs, and validated only for prediction of Metastasis-free survival (MFS) and recurrence-free interval (RFi) respectively. Hence besides PFS, MFS and RFI were also used as outcome indicator to compare the prognostic value of different models. For CNS SFTs with NTM, we compare the integrated model with WHO grade. The concordance index (C-index) and area under the curve of the receiver operating characteristic curve (AUC) were used to compare the performance of different risk models.

### Statistical analysis

SPSS statistical software version 17.0 and R software version 4.1.3 and graphpad prism 8.0 were used for all statistical analyses and figure plotting. PFS was estimated using the Kaplan–Meier method, and compared using the log-rank test. Univariate analyses using Cox proportional hazard regression models were used to test the significance of individual risk variables, and hazard ratios (HR) and 95% confidence interval (CI) were calculated. The R packages used in the current study included ComplexHeatmap v2.14.0, survival v3.4.0, ggpubr v0.6.0, ggplot2 v3.4.1, ggpubr v0.6.0, randomForestSRC v3.2.0, rms v6.5.0, Hmisc v4.8.0, pROC v1.18.0, ggalluvial v0.12.4, UpSetR v1.4.0. Chi-square test or fisher exact test were used to compare the difference within categorical data. Intergroup comparisons were performed using unpaired Wilcoxon rank-sum test. Kruskal–Wallis was used to compare the difference of multiple groups. All tests were two-sided and $p < 0.05$ was considered statistically significant.

### Reporting summary

Further information on research design is available in the Nature Portfolio Reporting Summary linked to this article.

## Data availability

Data used in the preparation of this manuscript are available within the Article, Supplementary Information, Supplementary Data and Source Data file. There are no restrictions on data access. Raw next generation sequencing data has been deposited in Genome Sequence Archive of National Genomics Data Center with open access (bioProject accession: PRJCA015954), according to Guidance of the Ministry of Science and Technology (MOST) for the Review and Approval of Human Genetic Resources https://bigd.big.ac.cn/gsa-human/browse/HRA004309. Raw imaging data files from immunohistochemistry and multiplex immunofluorescence staining are substantial and can be made available from the corresponding authors on request. Further information and requests for resources and reagents should be directed to and will be promptly fulfilled by the corresponding authors. Source data are provided with this paper.

## Code availability

The custom code used to analyze the images of multiplex immunofluorescence staining was published previously[38], and is available on GitHub (https://github.com/sorgerlab/cycif).

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

## Acknowledgements

This work was supported by the National Natural Science Foundation of China (82272931 to Z.D.), the Natural Science Foundation of Guangdong Province (2022A1515011197 to Z.D.), the Guangdong Science and Technology Department (2023B1212060013 to Q.L.).

## Author contributions

Z.D., Y.F., Q.L., and S.S. conceived of and designed the study. R.Z., Y.Y., C.H., Y.F., M.H., W.C., H.Z., P.M., Xiaojun Zhu, and G.S. provided the experimental materials and performed the clinical study. R.Z., W.C., D.L., W.Z., M.W., J.Y., Y.X., Xu Zhang, and L.S. performed the IHC and Sanger sequencing. R.Z., Y.Y., and C.H. performed risk modeling. X.Y. and W.Z. were responsible for bioinformatic data analysis. Yakang Long, B.X., J.P., S.W., and J.M. performed NGS experiments. R.Z., Y.Y., C.H., M.H., W.C., Fang Wang, D.W., Feng-Hua Wang, Yu-Hong Li, S.S., Q.L., Y.F., and Z.D. were responsible for data acquisition, analysis, and interpretation. R.Z. was responsible for drafting this manuscript. Z.D., Y.F., and Q.L. were responsible for the critical revision of the manuscript for important intellectual content. Z.D. contributed administrative support.

## Competing interests

The authors declare no competing interests.

## Additional information

¹State Key Laboratory of Oncology in South China, Sun Yat-Sen University Cancer Center, Guangzhou 510060, China. ²Department of Molecular Diagnostics, Sun Yat-sen University Cancer Center, Guangzhou 510060, China. ³Department of Pathology, The First Affiliated Hospital, Sun Yat-sen University, Guangzhou 510080, China. ⁴Department of Pathology, National Cancer Center/National Clinical Research Center for Cancer/Cancer Hospital, Chinese Academy of Medical Sciences and Peking Union Medical College, Beijing 100021, China. ⁵Department of Pathology, Sun Yat-sen University Cancer Center, Guangzhou 510060, China. ⁶Department of Musculoskeletal Oncology, Sun Yat-sen University Cancer Center, Guangzhou 510060, China. ⁷Second Department of Oncology, Guangdong Second Provincial General Hospital, Guangzhou 510317, China. ⁸Department of Medical Oncology, Sun Yat-sen University Cancer Center, Guangzhou 510060, China. ⁹Department of Pathology, Brigham and Women's Hospital, Harvard Medical School, Boston, MA 02115, USA. ¹⁰Guangdong Provincial Key Laboratory of Malignant Tumor Epigenetics and Gene Regulation, Guangdong-Hong Kong Joint Laboratory for RNA Medicine, Sun Yat-Sen Memorial Hospital, Sun Yat-Sen University, Guangzhou 510120, China. ¹¹Medical Research Center, Sun Yat-Sen Memorial Hospital, Sun Yat-Sen University, Guangzhou 510120, China. ¹²These authors contributed equally: Renjing Zhang, Yang Yang, Chunfang Hu, Mayan Huang, Wenjian Cen. ¹³These authors jointly supervised this work: Qin Li, Yan-Fen Feng, Ziming Du. ✉e-mail: liqin63@mail.sysu.edu.cn; fengyf@sysucc.org.cn; duzm1@sysucc.org.cn

