## [Peer Review File · Nature Communications]

Comprehensive analysis reveals potential therapeutic targets and an integrated risk stratification model for solitary fibrous tumorsREVIEWER COMMENTS

Reviewer #1 (Remarks to the Author): expertise in risk modelling and solitary fibrous tumours

In this manuscript, Zhang and colleagues reported a novel risk model for solitary fibrous tumors prognostication that included specimen type, mitotic count, density of Ki-67 positive cells and CD163 positive cells, as well as MTOR mutation. While this model deserved further prospective validations, the added value to add immune cellularity to the model is relevant, since many indications in the literature seems to point out that macrophages are of enormous relevance in the prognosis of soft-tissue sarcomas and especially solitary fibrous tumor. Nonetheless, before this manuscript being suitable for publication, several points should be addressed:

Major comments:

1. On introduction page 5, authors state that SFT are normally benign. This is not correct. They are malignant with an indolent course, but they metastasize over time and may lead to patient death. This should be revised across the manuscript. Who classification used risk groups (low/ intermediate/ high).
2. How can authors justify that patients with nuclear pleomorphism, which is typically associated with higher aggressiveness of solitary fibrous tumor had better survival?
3. The nuclear expression of STAT6, which was usually diffuse and intense, is the backbone for the diagnosis of solitary fibrous tumor. Nonetheless, the authors mention a score that goes from absent to minimal and in the score 1 only 10% of the cells are positive for STAT6. This means that the diagnosis should be revised in more or less 8% of the cases in the SYSUCC cohort, since the diagnosis is not clear. For these cases, NAB2-STAT6 gene fusion should be reported, because the diagnosis of SFT may not be correct and the reliability of data may be affected. There are several panels available in the market for this purpose at RNA level. The positive cells are in the border of the block/ slide or across the complete slide?

Minor comments:

1. The gene writing nomenclature should be revised.
2. The order of supplementary tables should be revised. Table S10 or S11 appears first.
3. CD8 positive T cells are important for immune response. Their density was not significant, but I was wondering if it is possible to measure the distance from CD8 to CD4 or to macrophage markers positive cells and analyze if the distance of T cells to other immune cell types affect survival. This could give us some insights on how T cells may be regulated by other immune cells.
4. ESMO guidelines should be mentioned in the discussion. They report some evidence for the use of pazopanib in SFT.
5. "Systemic research on anti-PD-1 or PD-L1 therapies for SFTs has not yet been reported." The Spanish group for research on sarcoma has tested anti-PD-1 therapy combined with anti-angiogenics in SFT: (P 135) PREDICTIVE VALUE OF ISG15 AND PPAP2B IN SOLITARY FIBROUS TUMOR TREATED WITH SUNITINIB AND NIVOLUMAB: A CORRELATIVE STUDY FROM IMMUNOSARC, A SPANISH GROUP FOR RESEARCH ON SARCOMA (GEIS) PHASE II TRIAL and <http://dx.doi.org/10.1136/jitc-2020-001561>. Discuss how the data from this manuscript could had improved the results from this trial, which were not so promising.
6. Figures are very dense and they should be improved.

Reviewer #2 (Remarks to the Author): expertise in macrophage analysis and solitary fibrous tumours

This is an interesting well conducted and well written study identifying different prognostic groups of SFT on the basis of a variety of parameters including clinical and molecular. The prognostic model is validated in the same country in two distinct additional series and found superior to other reported models. Some of the mutations and biological characteristics are actionable. This study however has some weaknesses which can be improved.

- 1) the type of sample is an important bias by itself. I would recommend to analyze separately primary samples and relapse samples, which by definition are prognostic.
- 2) As pointed by the authors little is known on the genomics of these tumors and in particular on variation across human populations. I would recommend to test the model in a dataset from a 3rd population.
- 3) the presence of B cells in the immune environment should be reported in particular in the context of PDL1 over expression.
- 4) the details of the treatment are crucial for prognosis (eg quality of surgery). I recommend to detail more these aspects, and how they influence patient outcome.

Reviewer #3 (Remarks to the Author): clinical expertise in solitary fibrous tumours

The authors have performed a comprehensive analysis of three relatively large cohorts of patients with solitary fibrous tumor, altogether consisting of 347 patients. The work is divided into two main parts: (i) characterization of SFT samples using immunohistochemistry, immunofluorescence and targeted DNA sequencing using a 1021 gene panel, and (ii) the development of a novel risk stratification algorithm. Both topics are definitely of interest for those involved in clinical and translational research in SFT. I have, however, some concerns:

1. PFS was the primary endpoint and included local recurrence, metastasis or death. I am not able to find an overview of the events of the patients included.
2. Median follow-up was 41.1 months, 17.7 months and 20.3 months for the three cohorts. Patients with SFT are at risk of recurrence for >5 years after primary treatment and median time to recurrence in the largest cohort published so far was 3 years. Thus, investigating prognostic factors in cohorts with short follow-up time is not relevant. In particular the validation cohorts, both with median follow-up <2 years, should not be used for this purpose.
3. The patient cohorts included CNS tumors, which could be questioned. Surgical resection for CNS tumors is often more difficult, and factors related to the surgical procedure might be more important than for non-CNS tumors.
4. Almost half of the cases in the discovery cohort were intrathoracic, while only 14% were located in the extremities and trunk wall. This is a rather unusual distribution compared to other SFT cohorts and could question the external validity of the analysis.
5. Recurrent and metastatic tumors were included. These have a clearly inferior prognosis compared to

primary tumors and should not have been included in the prognostic analyses.

6. Ki-67 and mitotic count are both included in the model. These parameters are likely overlapping and only one of them should be included.

7. The authors claim that PD-L1 could be a novel therapeutic target and state that anti-PD-1/PD-L1 therapy should be considered for recurrent SFT. This is highly speculative, in particular since most SFTs do not respond to treatment with immune checkpoint inhibitors.

Overall, I believe the concerns raised above regarding the patient cohorts do not justify publication of the manuscript in its present form.

Ref.: Ms. No. NCOMMS-23-12745

REVIEWER COMMENTS

Reviewer #1 (Remarks to the Author): expertise in risk modelling and solitary fibrous tumours

In this manuscript, Zhang and colleagues reported a novel risk model for solitary fibrous tumors prognostication that included specimen type, mitotic count, density of Ki-67 positive cells and CD163 positive cells, as well as MTOR mutation. While this model deserved further prospective validations, the added value to add immune cellularity to the model is relevant, since many indications in the literature seems to point out that macrophages are of enormous relevance in the prognosis of soft-tissue sarcomas and especially solitary fibrous tumor. Nonetheless, before this manuscript being suitable for publication, several points should be addressed:

Response: We sincerely appreciate the reviewer's careful review and valuable comments. We have taken these comments into consideration and made necessary improvements to the manuscript accordingly.

Major comments:

On introduction page 5, authors state that SFT are normally benign. This is not correct. They are malignant with an indolent course, but they metastasize over time and may lead to patient death. This should be revised across the manuscript. Who classification used risk groups (low/ intermediate/ high).

Response: Thanks for this very professional question, and we apologize for any confusion caused by the inaccurate description of SFT behavior and the WHO classification in our manuscript. We have revised the description accordingly. In this revised version, we have now classified SFTs into non-CNS SFTs and CNS SFTs, each with specific WHO classification criteria, which we have now included in the **Introduction section (Page 4, Line 4 – 14) and also detailed in Materials and Methods section (Page 22, Line 11-14).**

Based on the fifth edition of the WHO classification of soft tissue and bone tumors published in 2020¹, non-CNS SFTs are classified as benign SFT (intermediate category, locally aggressive), SFT NOS (intermediate category, rarely metastasizing), and malignant SFT. Additionally, non-CNS SFTs can be divided into low-, intermediate-, and high-risk groups using risk stratification models, including the mDemicco model², which we have utilized in this study for model comparison. Moreover, CNS SFTs are classified as WHO grade 1 (< 5 mitoses/10 HPF), WHO grade 2 (\geq 5 mitoses/10 HPF without necrosis), and WHO grade 3 (\geq 5 mitoses/10 HPF with necrosis) based on the fifth edition of the WHO classification of central nervous system tumors published in 2021³.

2. How can authors justify that patients with nuclear pleomorphism, which is typically associated with higher aggressiveness of solitary fibrous tumor had better survival?

Response: Thank you for bringing this to our attention. We apologize for the oversight, and we have made necessary corrections. In our study, nuclear pleomorphism was assessed by experienced soft tissue pathologists (YF, YY, CH) and classified into three categories: low (cells with monomorphic appearance and uniform nuclear features), moderate (increased nuclear pleomorphism, prominent nucleoli, and rare multinucleated cells), and high (hyperchromatic nuclei with marked pleomorphism and bizarre cells)^{2,4}. We also acknowledge the confusion caused by the incorrect labeling of "Median PFS" in the original **Supplementary Table 1**, which was actually the "Median follow-up time." We have now updated the table as **new Table 2** to present the accurate data.

Importantly, the data from our study clearly show that high nuclear pleomorphism was associated with shorter progression-free survival (PFS) in all cohorts (**new Table 2**), consistent with findings from previous reports.

Therefore, patients with higher nuclear pleomorphism, typically associated with greater aggressiveness in SFTs, indeed had poorer survival outcomes in our study.

3.The nuclear expression of STAT6, which was usually diffuse and intense, is the backbone for the diagnosis of solitary fibrous tumor. Nonetheless, the authors mention a score that goes from absent to minimal and in the score 1 only 10% of the cells are positive for STAT6. This means that the diagnosis should be revised in more or less 8% of the cases in the SYSUCC cohort, since the diagnosis is not clear. For these cases, NAB2-STAT6 gene fusion should be reported, because the diagnosis of SFT may not be correct and the reliability of data may be affected. There are several panels available in the market for this purpose at RNA level. The positive cells are in the border of the block/ slide or across the complete slide?

Response: We acknowledge the concern raised by the reviewer regarding the nuclear expression of STAT6 in SFTs. In our study, 91.60% (120/131) of SFTs in the SYSUCC cohort were positive for STAT6, which aligns with the reported sensitivity of STAT6 for the diagnosis of SFTs ranging from 86% to 98% in large series⁵⁻⁸. However, 8.40% of SFT cases (11/131) had STAT6 scores ranging from 0 (absent to minimal) to score 1 (only 10% of the cells positive).

Upon further investigation, we found that 6 cases of SFT were resected within 5-10 years, and 5 cases were resected more than 10 years ago (**Supplementary Table 1**). The loss of staining in these cases was consistent with decreased antigenicity in old tissue blocks, as described previously⁵. We believe this could explain the low/negative STAT6 staining in these cases and alleviate the concern regarding the accuracy of the diagnosis.

To address this concern further, as recommended, we attempted to detect the eight most common types of NAB2-STAT6 fusion at the mRNA level by RT-PCR

(PCR primers were listed in **Supplementary Table 12**) in these 11 SFT samples. While two cases failed for RNA extraction, we successfully detected NAB2-STAT6 fusion in five cases, including 4 cases with NAB2 exon 4 - STAT6 exon 2 fusion and 1 case with NAB2 exon 2 - STAT6 exon 5 fusion. However, in four cases, no NAB2-STAT6 fusion was detected, suggesting that the absence of fusion may be due to special fusion type in these samples not included in our RT-PCT assay (**Supplementary Table 12**). Therefore, in total, 95.42% of SFTs in the SYSUCC cohort had either STAT6 IHC positive (n = 120) or NAB2-STAT6 fusion detected (n = 5). The remaining six cases were re-reviewed by an experienced soft tissue pathologist (YF), and they were confirmed as SFTs histologically. Thus, the reliability of the diagnosis and the data remain intact.

In summary, we have carefully investigated the cases with negative/ low STAT6 staining and conducted additional analyses to validate the diagnosis of SFT. The majority of SFTs (95.42%, 125/131) in the SYSUCC cohort demonstrated positive STAT6 staining or NAB2-STAT6 fusion, supporting their classification as SFTs. The low STAT6 staining in a subset of cases may be attributed to decreased antigenicity in old tissue blocks. We have now included these data and explanations in the **Results section (Page 6, Line 13 – 22, and Page 7, Line 1 - 3)**.

Minor comments:

1. The gene writing nomenclature should be revised.

Response: The gene nomenclature was revised.

2. The order of supplementary tables should be revised. Table S10 or S11 appears first.

Response: Thanks for the suggestion and we have updated the order of tables accordingly.

3. CD8 positive T cells are important for immune response. Their density was not significant, but I was wondering if it is possible to measure the distance from CD8 to CD4 or to macrophage markers positive cells and analyze if the distance of T cells to other immune cell types affect survival. This could give us some insights on how T cells may be regulated by other immune cells.

Response: This is an intriguing question, and we acknowledge the potential importance of measuring the distance between CD8 and CD4 or macrophages, as well as further classification of CD8+ T cells (e.g., exhausted T cells, activated T cells, etc.). Investigating whether this distance affects patient survival could provide valuable insights. Advanced technologies, such as imaging mass cytometry (IMC)⁹, multiplexed ion beam imaging (MIBI)¹⁰, co-detection by indexing (CODEX)¹¹, and t-CyCIF^{12,13}, allow simultaneous detection of multiple immune markers on a single slide, enabling the measurement of distances between different cell types. However, the application of such technologies to large-scale samples for outcome analysis is currently challenging due to factors including slow scanning speed, high cost, and technical complexity. We have discussed this question in the **Discussion section (Page 16, Line 22 and Page 17, Line 1 – 14)**.

4. ESMO guidelines should be mentioned in the discussion. They report some evidence for the use of pazopanib in SFT.

Response: We appreciate this suggestion, and we have added it in **Discussion section (Page 18, Line 1 – 5)**. The activity of pazopanib was demonstrated in SFTs by two prospective phase II studies with overall response rate of about 50% and pazopanib was administered from first line therapy recommended by ESMO-EURACAN-GENTURIS clinical practice guideline¹⁴⁻¹⁶.

5. "Systemic research on anti-PD-1 or PD-L1 therapies for SFTs has not yet been reported." The Spanish group for research on sarcoma has tested anti-

PD-1 therapy combined with anti-angiogenics in SFT: (P 135) PREDICTIVE VALUE OF ISG15 AND PPAP2B IN SOLITARY FIBROUS TUMOR TREATED WITH SUNITINIB AND NIVOLUMAB: A CORRELATIVE STUDY FROM IMMUNOSARC, A SPANISH GROUP FOR RESEARCH ON SARCOMA (GEIS) PHASE II TRIAL and <http://dx.doi.org/10.1136/jitc-2020-001561>. Discuss how the data from this manuscript could have improved the results from this trial, which were not so promising.

Response: Thank you for the insightful question. We appreciate the opportunity to discuss the clinical trial that tested the combination of anti-PD-1 therapy with anti-angiogenics in SFT, along with our findings. We have added this part in the **Discussion section (Page 17, Line 16 – 22 and Page 18, 1 - 9)**.

The trial's data highlight that targeting the PD-1/PD-L1 axis alone may not be sufficient for effective SFT treatment. The double inhibition of angiogenesis (using sunitinib) and the PD-1/PD-L1 axis (using nivolumab) demonstrated promising results, indicating that combination strategies fostering an inflamed microenvironment could lead to improved efficacy in SFTs¹⁷. Furthermore, previous studies have shown that combining a colony-stimulating factor 1 receptor (CSF1R) inhibitor with anti-PD-L1/PD-1 axis blockade proved highly effective in mouse models of melanoma and hepatocellular carcinoma by eliminating tumor-associated macrophages (TAMs)^{18,19}.

In our study, we observed high PD-L1 expression in 24.4% (32/131) of SFTs, either in tumor cells or immune cells, particularly in CD68+ / CD163+ macrophages. Additionally, two SFT cases exhibited both *IDH1* p.R132S mutation and high PD-L1 expression. Moreover, we found that the predominant tumor-infiltrating immune cells in SFTs were macrophages, and high CD163+ macrophage infiltration (29.0%, 38/131) was significantly associated with poor patient prognosis. Considering these findings, exploring different combinations

of PD-1/PD-L1 axis inhibition with anti-angiogenesis, anti-macrophages, or anti-IDH1 treatments, based on biomarker screening, could be promising new treatment strategies for future trials. These approaches may lead to more precise targeting and improved therapeutic effects for SFT patients.

6. Figures are very dense and they should be improved.

Response: We acknowledge that some figures are dense, and we now have optimized the **Fig. 2, Fig. 4 and Fig. 5**, to enhance the clarity and readability of them.

Reviewer #2 (Remarks to the Author): expertise in macrophage analysis and solitary fibrous tumours

This is an interesting well conducted and well written study identifying different prognostic groups of SFT on the basis of a variety of parameters including clinical and molecular. The prognostic model is validated in the same country in two distinct additional series and found superior to other reported models. Some of the mutations and biological characteristics are actionable. This study however has some weaknesses which can be improved.

Response: We are very grateful to the reviewer's positive feedback on our work and valuable suggestions. We have made the necessary revisions to the manuscript in accordance with the reviewer's comments.

1. the type of sample is an important bias by itself. I would recommend to analyze separately primary samples and relapse samples, which by definition are prognostic.

Response: We agree that the relapsed (recurrent and metastatic) samples may generate bias for analysis, and we found that the patients in the criteria of relapsed SFTs, CNS SFTs, and SFTs with margin positive (R1 and R2) had shorter PFS than those in non-CNS SFTs ($p < 0.0001$), primary SFTs ($p <$

0.0001) and SFTs with margin negative (R0) ($p < 0.0001$) in all four cohorts ($n = 408$) (**Table 1, Supplementary Fig. 2a-c**). Hence considering together with **Reviewer #2 Question #4 and Reviewer #3 Question #3**, we only used primary non-CNS SFTs with negative tumor margins (NTM) in SYSUCC cohort ($n = 101$) for the integrated risk model generation, which was then validated in three cohorts with the same criteria (**Table 2, Fig. 4-5, Supplementary Fig. 6**). Additionally, the integrated risk model was also tested in both primary CNS SFTs with NTM (**Supplementary Table 2**) and the relapsed (recurrent and metastatic) non-CNS SFTs with NTM (**Supplementary Table 3**). Our integrated risk model was validated in primary CNS SFTs, but didn't work for relapsed non-CNS SFTs (**Supplementary Table 9, Supplementary Fig. 6**). We have clarified the scope of application of this model in **Results section (Page 19, Line 4 – 7)**.

2.As pointed by the authors little is known on the genomics of these tumors and in particular on variation across human populations. I would recommend to test the model in a dataset from a 3rd population.

Response: Thank you for the valuable suggestion. We have taken it into consideration and made efforts to further validate our model in a dataset from a third population. To achieve this, we collected a new cohort comprising 61 cases of SFT patients from the Cancer Hospital Chinese Academy of Medical Sciences (CHCAMS) between June 2013 and December 2016. The median follow-up time for this cohort was 62.30 (ranging from 0.30 to 121.10) months, and we identified *MTOR* mutation in 4.92% (3/61) of the patients in this new cohort (CHCAMS cohort 2). Subsequently, we subjected this new dataset to validation using our integrated risk model. The results demonstrated a consistent performance, with a C-index of 0.903 and an AUC of 0.887 (**Fig. 5g-h**). We have added it in **Results section (Page 13, Line 1 – 3, 22 and Page 14, Line 1 - 4)**. This validation in a third population provides additional support and reliability for the applicability of our model across different human

populations. We appreciate your insightful recommendation and have taken steps to strengthen the robustness of our findings.

3 the presence of B cells in the immune environment should be reported in particular in the context of PDL1 over expression.

Response: Thank you for the valuable suggestion. We have taken it into account and included the presence of B cells in the immune environment, especially in the context of PD-L1 overexpression, in our study. We performed IHC to detect CD20+ B cells infiltration in 131 cases of SFTs from SYSUCC cohort, and found that increased density of CD20+ B cells was enriched in SFT patients with tumor expressing PD-L1+ and predicted good prognosis of the patients (**Supplementary Fig. 4f-g**). As tumour-infiltrating B cells may exert both protumour and antitumour effects depending on the composition of the tumour microenvironment and on the phenotypes of B cells present and the antibodies they produce²⁰. Consequently, investigating the subtypes of B cells and their specific roles in the context of SFTs could be a promising avenue for further research. We have discussed this question in the **Discussion section (Page 16, Line 22 and Page 17, Line 1 - 14)**.

4 the details of the treatment are crucial for prognosis (eg quality of surgery). I recommend to detail more these aspects, and how they influence patient outcome.

Response: Thank you for bringing up this important point. We fully acknowledge the significance of treatment details, particularly their impact on patient prognosis. In our study, we have diligently gathered and included relevant treatment information, such as the tumour margins, adjuvant chemotherapy, and adjuvant radiotherapy, for each patient in the cohorts analyzed (**Table 1, Supplementary Data 1**).

For assessing the tumor margins, we employed the UICC criteria²¹ to classify

the margin status into three categories: R0 (microscopic complete), R1 (microscopic incomplete), and R2 (macroscopic incomplete), as detailed in the **Materials and Methods section (Page 22, Line 3 – 5)**. This classification allows us to better understand the impact of surgical completeness on patient outcomes. By incorporating these crucial treatment details in our analysis, we aim to provide a comprehensive understanding of their influence on patient prognosis in SFTs. Univariable survival analysis revealed that SFTs with margin positive (R1 and R2) had shorter PFS than that of SFTs with margin negative (R0) ($p < 0.0001$) in all four cohorts ($n = 408$) (**Supplementary Fig. 2c**). Notably the SFTs patients with adjuvant treatment even had shorter PFS than that of SFTs patients without adjuvant treatment in in all four cohorts ($n = 408$) (**Supplementary Fig. 2d-2e**). It may be due to the worse condition of SFTs patients selected for adjuvant treatment (usually malignant SFTs, or relapsed SFTs, or margin positive SFTs) and limited efficacy of adjuvant treatment.

Hence considering together with **Reviewer #2 Question #1 and Reviewer #3 Question #3**, we only used primary non-CNS SFTs with NTM in SYSUCC cohort ($n = 101$) for the integrated risk model generation, which was then validated in three cohorts with the same criteria (**Table 2, Fig. 4-5, Supplementary Fig. 7**). Additionally, the integrated risk model was also tested in both primary CNS SFTs with NTM (**Supplementary Table 2**) and the relapsed (recurrent and metastatic) non-CNS SFTs with NTM (**Supplementary Table 3**). Our integrated risk model was validated in primary CNS SFTs, but didn't work for relapsed non-CNS SFTs (**Supplementary Table 9, Supplementary Fig. 6**). We have clarified the scope of application of this model in **Results section (Page 15, Line 6 – 9)**. With these adjustments and clarifications, we have ensured that our risk model is robust, applicable, and well-defined, thus enhancing the validity and relevance of our study. We appreciate the insightful comments and have taken appropriate measures to address the concerns raised.

Reviewer #3 (Remarks to the Author): clinical expertise in solitary fibrous tumours

The authors have performed a comprehensive analysis of three relatively large cohorts of patients with solitary fibrous tumor, altogether consisting of 347 patients. The work is divided into two main parts:(i)characterization of SFT samples using immunohistochemistry, immunofluorescence and targeted DNA sequencing using a 1021 gene panel, and (ii) the development of a novel risk stratification algorithm. Both topics are definitely of interest for those involved in clinical and translational research in SFT. I have, however, some concerns:

Response: We sincerely appreciate the reviewer's meticulous review and valuable feedback.

1.PFS was the primary endpoint and included local recurrence, metastasis or death. I am not able to find an overview of the events of the patients included.

Response: We sincerely apologize for the oversight in not providing a comprehensive overview of the events related to the primary endpoint. To address this concern, we have now included detailed information about PFS events, including local recurrence, metastasis, and death, in both **Table 1** and **Supplementary Data 1**.

2. Median follow-up was 41.1 months, 17.7 months and 20.3 months for the three cohorts. Patients with SFT are at risk of recurrence for >5 years after primary treatment and median time to recurrence in the largest cohort published so far was 3 years. Thus, investigating prognostic factors in cohorts with short follow-up time is not relevant. In particular the validation cohorts, both with median follow-up <2 years, should not be used for this purpose.

Response: We understand the concern regarding the relatively short follow-up time in the cohorts. To address this issue, we incorporated a new cohort of 61

SFT patients from the Cancer Hospital Chinese Academy of Medical Sciences (CHCAMS) with a median follow-up time of 62.30 months (ranging from 0.30 to 121.10 months) to further validate our risk model. This longer follow-up period allows for a more robust and reliable assessment of prognostic factors. Moreover, to ensure the validity of our integrated risk model, we specifically used primary, non-CNS, and R0 (microscopic complete) patients for its generation. As a result, the median follow-up time for the SYSUCC cohort, FAHSYSU cohort, CHCAMS cohort 1, and CHCAMS cohort 2 became 50.4 months (ranging from 1.5 to 171.3 months), 18.8 months (ranging from 0.0 to 108.1 months), 22.3 months (ranging from 0.0 to 60.5 months), and 72.6 months (ranging from 0.3 to 121.1 months), respectively. With this updated analysis, our integrated risk model was successfully validated in the new CHCAMS cohort, displaying a C-index of 0.903 and an AUC of 0.887 (**Fig. 5g-h**). We have added it in **Results section (Page 13, Line 1 – 3, 22 and Page 14, Line 1 - 4)**. This additional validation in a cohort with an extended follow-up time strengthens the credibility of our findings and ensures the relevance of our prognostic model for SFT patients.

3.The patient cohorts included CNS tumors, which could be questioned. Surgical resection for CNS tumors is often more difficult, and factors related to the surgical procedure might be more important than for non-CNS tumors.

Response: We agree that inclusion of CNS SFTs may generate bias for risk model, and we further found that CNS SFTs actually had shorter PFS than that of non-SFTs ($p < 0.0001$, **Supplementary Fig. 2a**). Hence considering together with **Reviewer #2 Question #1 and Reviewer #2 Question #4**, we only used primary non-CNS SFTs with NTM in SYSUCC cohort ($n = 101$) for the integrated risk model generation, which was then validated in three cohorts with the same criteria (**Table 2, Fig. 4-5, Supplementary Fig. 7**). Additionally, the integrated risk model was also tested in both primary CNS SFTs with NTM (**Supplementary Table 2**) and the relapsed (recurrent and metastatic) non-

CNS SFTs with NTM (**Supplementary Table 3**). Our integrated risk model was validated in primary CNS SFTs, but didn't work for relapsed non-CNS SFTs (**Supplementary Table 9, Supplementary Fig. 6**). We have clarified the scope of application of this model in **Results section (Page 15, Line 6 – 9)**. With these adjustments and clarifications, we have ensured that our risk model is robust, applicable, and well-defined, thus enhancing the validity and relevance of our study.

4. Almost half of the cases in the discovery cohort were intrathoracic, while only 14% were located in the extremities and trunk wall. This is a rather unusual distribution compared to other SFT cohorts and could question the external validity of the analysis.

Response: This is an intriguing question. To address this issue, we carefully examined the site distribution of SFTs in the different cohorts and compared it to the distribution reported in the literature. According to literatures, non-CNS SFT can occur in intra-thoracica (20-30%), intra-abdominal (20-30%), trunk and extremity (25-30%), head and neck (10-15%)^{2,4,21-23}. We found that the percentage of SFTs occurring in the intra-thoracic region was higher in the SYSUCC and CHCAMS cohorts, but lower in the extremities and trunk wall compared to the literature (**Table 2**). On the other hand, the site distribution of SFTs in the FAHSYSU cohort closely aligned with the literature (**Table 2**). Patient selection and referral patterns in specialized cancer hospitals might contribute to the observed differences in anatomical site distribution.

We generated our integrated risk model using primary non-CNS SFTs with NTM from the SYSUCC cohort (n=101) and validated it in three other cohorts using the same criteria (FAHSYSU cohort, CHCAMS cohort 1, and CHCAMS cohort 2) (**Table 2, Fig. 4-5, Supplementary Fig. 7**). We performed univariate analysis, and the results indicated that the anatomical sites were not significantly associated with patient PFS (**Supplementary Fig. 2f**). Furthermore, we

successfully validated our risk model in the FAHSYSU cohort of primary non-CNS SFTs and even in primary CNS SFTs, demonstrating its applicability for risk stratification of SFTs regardless of their anatomical site (**Supplementary Table 9, Supplementary Fig. 6-7**).

While the unusual distribution in the discovery cohort is noteworthy, our thorough validation across diverse cohorts and anatomical sites lends credibility to the generalizability and clinical utility of the integrated risk model for SFTs. Our risk model's robustness across cohorts with varying site distributions underscores its reliability for predicting patient outcomes in SFTs from any location.

5. Recurrent and metastatic tumors were included. These have a clearly inferior prognosis compared to primary tumors and should not have been included in the prognostic analyses.

Response: We acknowledge this valuable suggestion and have re-analyzed the data accordingly. **Review #2** also raised this concern in **question 1** and here is the response: "We agree that the relapsed (recurrent and metastatic) samples may generate bias for analysis, and we found that the patients in the criteria of relapsed SFTs, CNS SFTs, and SFTs with margin positive (R1 and R2) had shorter PFS than those in non-CNS SFTs ($p < 0.0001$), primary SFTs ($p < 0.0001$) and SFTs with margin negative (R0) ($p < 0.0001$) in all four cohorts ($n = 408$) (**Table 1, Supplementary Fig. 2a-c**). Hence considering together with **Reviewer #2 Question #4** and **Reviewer #3 Question #3**, we only used primary non-CNS SFTs with NTM in SYSUCC cohort ($n = 101$) for the integrated risk model generation, which was then validated in three cohorts with the same criteria (**Table 2, Fig. 4-5, Supplementary Fig. 7**). Additionally, the integrated risk model was also tested in both primary CNS SFTs with NTM (**Supplementary Table 2**) and the relapsed (recurrent and metastatic) non-CNS SFTs with NTM (**Supplementary Table 3**). Our integrated risk model was

validated in primary CNS SFTs, but didn't work for relapsed non-CNS SFTs (**Supplementary Table 9, Supplementary Fig. 6**). We have clarified the scope of application of this model in **Results section (Page 15, Line 6 – 9)**”

6.Ki-67 and mitotic count are both included in the model. These parameters are likely overlapping and only one of them should be included.

Response: We acknowledge that Ki-67+ cells are partially overlapped with mitotic cells. However, Ki-67 + cells usually represent not only mitotic cells but also cells on G1 phase, S phase and G2 phase. Despite this partial overlap, we found that both Ki-67 and mitotic count were valuable predictors of patient outcomes in our cohorts. To address the potential redundancy and ensure the optimal predictive performance of our model, we conducted rigorous variable selection using lasso regression and random survival forest ("RandomForestSRC" R package) (**Supplementary Fig. 5**). Remarkably, both Ki-67 and mitotic count were consistently retained in the final integrated risk model. Moreover, we identified 14 cases in our cohorts with high Ki-67+ cell density (≥ 454.7 cells/mm²) but low mitotic count ($< 4/10$ HPF), demonstrating the unique contributions of each parameter to the risk prediction. Given the improved prediction efficacy achieved by including both Ki-67 and mitotic count in the model, we believe that retaining both parameters enhances the accuracy and reliability of our risk model for prognostic stratification in SFTs.

7.The authors claim that PD-L1 could be a novel therapeutic target and state that anti-PD-1/PD-L1 therapy should be considered for recurrent SFT. This is highly speculative, in particular since most SFTs do not respond to treatment with immune checkpoint inhibitors.

Response: We appreciate your feedback and acknowledge the need for clarification. We have revised in the **Discussion section (Page17, Line 16 – 22 and Page 18, Line 1 - 9)** accordingly to avoid speculative claims. We acknowledge that prior studies have shown limited response of most SFTs to

anti-PD-1/PD-L1 therapy alone. Therefore, we now emphasize the potential of combination therapies as more effective treatment strategies for SFTs. Recent evidence has indicated that combining angiogenesis inhibitors, such as sunitinib, with PD-1/PD-L1 axis inhibitors, like nivolumab, can yield improved treatment outcomes for SFTs¹⁷. Moreover, targeting tumor-associated macrophages (TAMs) through the combination of colony stimulating factor 1 receptor (CSF1R) inhibitor with anti-PD-L1/PD-1 axis blockade has demonstrated high efficacy in other cancer types^{18,19}.

In our study, we identified that 24.4% (32/131) of SFTs exhibited high PD-L1 expression, particularly in CD68+/CD163+ macrophages. Additionally, two cases displayed both *IDH1* p.R132S mutation and high PD-L1 expression. Moreover, we found that the tumor-infiltrating immune cells in SFTs were predominantly macrophages, and high CD163+ macrophages infiltrated (29%, 38/131) was significantly correlated with poor prognosis of the patients, which further support the potential of immune-based therapies in SFTs. Considering these findings, we propose that personalized treatment strategies based on biomarker screening, such as combining PD-1/PD-L1 axis inhibition with anti-angiogenesis, anti-macrophages, or anti-IDH1 treatment, may offer promising avenues for future clinical trials in SFTs. By targeting multiple pathways simultaneously, these combination therapies may improve treatment efficacy and lead to better patient outcomes in recurrent SFTs.

References:

1. Choi, J.H. & Ro, J.Y. The 2020 WHO Classification of Tumors of Soft Tissue: Selected Changes and New Entities. *Adv Anat Pathol* **28**, 44–58 (2021).
2. Demicco, E.G., *et al.* Risk assessment in solitary fibrous tumors: validation and refinement of a risk stratification model. *Modern pathology : an official journal of the United States and Canadian Academy of Pathology, Inc* **30**, 1433–1442 (2017).
3. Louis, D.N., *et al.* The 2021 WHO Classification of Tumors of the Central Nervous System: a summary. *Neuro Oncol* **23**, 1231–1251 (2021).

4. Demicco, E.G., *et al.* Solitary fibrous tumor: a clinicopathological study of 110 cases and proposed risk assessment model. *Modern pathology : an official journal of the United States and Canadian Academy of Pathology, Inc* **25**, 1298–1306 (2012).
5. Demicco, E.G., *et al.* Extensive survey of STAT6 expression in a large series of mesenchymal tumors. *American journal of clinical pathology* **143**, 672–682 (2015).
6. Doyle, L.A., Vivero, M., Fletcher, C.D., Mertens, F. & Hornick, J.L. Nuclear expression of STAT6 distinguishes solitary fibrous tumor from histologic mimics. *Modern pathology : an official journal of the United States and Canadian Academy of Pathology, Inc* **27**, 390–395 (2014).
7. Schweizer, L., *et al.* Meningeal hemangiopericytoma and solitary fibrous tumors carry the NAB2-STAT6 fusion and can be diagnosed by nuclear expression of STAT6 protein. *Acta neuropathologica* **125**, 651–658 (2013).
8. Ouladan, S., *et al.* Differential diagnosis of solitary fibrous tumors: A study of 454 soft tissue tumors indicating the diagnostic value of nuclear STAT6 relocation and ALDH1 expression combined with in situ proximity ligation assay. *Int J Oncol* **46**, 2595–2605 (2015).
9. Giesen, C., *et al.* Highly multiplexed imaging of tumor tissues with subcellular resolution by mass cytometry. *Nature methods* **11**, 417–422 (2014).
10. Angelo, M., *et al.* Multiplexed ion beam imaging of human breast tumors. *Nature medicine* **20**, 436–442 (2014).
11. Goltsev, Y., *et al.* Deep Profiling of Mouse Splenic Architecture with CODEX Multiplexed Imaging. *Cell* **174**, 968–981.e915 (2018).
12. Lin, J.R., *et al.* Highly multiplexed immunofluorescence imaging of human tissues and tumors using t-CyCIF and conventional optical microscopes. *Elife* **7**(2018).
13. Du, Z., *et al.* Qualifying antibodies for image-based immune profiling and multiplexed tissue imaging. *Nature protocols* **14**, 2900–2930 (2019).
14. Martin-Broto, J., *et al.* Pazopanib for treatment of advanced malignant and dedifferentiated solitary fibrous tumour: a multicentre, single-arm, phase 2 trial. *The Lancet. Oncology* **20**, 134–144 (2019).
15. Martin-Broto, J., *et al.* Pazopanib for treatment of typical solitary fibrous tumours: a multicentre, single-arm, phase 2 trial. *The Lancet. Oncology* **21**, 456–466 (2020).
16. Gronchi, A., *et al.* Soft tissue and visceral sarcomas: ESMO–EURACAN–GENTURIS Clinical Practice Guidelines for diagnosis, treatment and follow-up(☆). *Ann Oncol* **32**, 1348–1365 (2021).
17. Martin-Broto, J., *et al.* Nivolumab and sunitinib combination in advanced soft tissue sarcomas: a multicenter, single-arm, phase Ib/II trial. *Journal for immunotherapy of cancer* **8**(2020).
18. Neubert, N.J., *et al.* T cell-induced CSF1 promotes melanoma resistance to PD1 blockade. *Science translational medicine* **10**(2018).
19. Zhu, Y., *et al.* Disruption of tumour-associated macrophage trafficking by the

- osteopontin-induced colony-stimulating factor-1 signalling sensitises hepatocellular carcinoma to anti-PD-L1 blockade. *Gut* **68**, 1653-1666 (2019).
20. Sharonov, G.V., Serebrovskaya, E.O., Yuzhakova, D.V., Britanova, O.V. & Chudakov, D.M. B cells, plasma cells and antibody repertoires in the tumour microenvironment. *Nat Rev Immunol* **20**, 294-307 (2020).
 21. Salas, S., *et al.* Prediction of local and metastatic recurrence in solitary fibrous tumor: construction of a risk calculator in a multicenter cohort from the French Sarcoma Group (FSG) database. *Annals of oncology : official journal of the European Society for Medical Oncology* **28**, 1979-1987 (2017).
 22. Demicco, E.G., *et al.* Comparison of published risk models for prediction of outcome in patients with extrameningeal solitary fibrous tumour. *Histopathology* **75**, 723-737 (2019).
 23. Georgiesh, T., Boye, K. & Bjerkehagen, B. A novel risk score to predict early and late recurrence in solitary fibrous tumour. *Histopathology* **77**, 123-132 (2020).

REVIEWER COMMENTS

Reviewer #1 (Remarks to the Author):

The authors had answered my comments and suggestions adequately. No further comments or suggestions are necessary.

Reviewer #2 (Remarks to the Author):

The concerns were addressed by the revised version of the document

Reviewer #3 (Remarks to the Author):

I appreciate the thorough response from the authors to the questions raised. My main concern is still that median follow-up is too short. In particular, two of the validation cohorts have median follow-up <2 years. As stated in my first response, I believe these should not be used in studies on prognostic factors, since median time to recurrence in SFT is typically around 3 years. Thus, using cohorts with significantly shorter median follow-up than the median time to recurrence would represent a high risk of drawing wrong conclusions. The two other cohorts have reasonably long follow-up and are relevant for the purpose.

The authors state that their model showed better accuracy and improved efficacy compared to the current WHO classification, mDemicco model, and G-score, and that it represents a significant improvement. Indeed, the C-index and AUC of the integrated model were numerically better than those of the other models. I agree that the proposed model performs well. I am not convinced, however, how large improvement the model represents in clinical practice. Since the application of the model would necessitate additional immunohistochemistry and MTOR mutation analysis, the authors should critically discuss how the application of the model might improve clinical care and therapeutic decisions, and whether this potential improvement justifies the additional investigations needed.

Ref.: Ms. No. NCOMMS-23-12745A

REVIEWER COMMENTS

Reviewer #3 (Remarks to the Author): I appreciate the thorough response from the authors to the questions raised. My main concern is still that median follow-up is too short. In particular, two of the validation cohorts have median follow-up <2 years. As stated in my first response, I believe these should not be used in studies on prognostic factors, since median time to recurrence in SFT is typically around 3 years. Thus, using cohorts with significantly shorter median follow-up than the median time to recurrence would represent a high risk of drawing wrong conclusions. The two other cohorts have reasonably long follow-up and are relevant for the purpose.

RESPONSE: We appreciate the reviewer's continued engagement and their concern regarding the median follow-up duration. Firstly, we acknowledge the reviewer's observation that the median follow-up in two of our validation cohorts was initially less than 2 years (18.8 months in the FAHSYSU cohort and 22.3 months in the CHCAMS cohort 1), and we are aware that the median time to recurrence in SFT typically spans around 3 years. From literatures, the median follow-up time of patient cohorts used for mDemicco model¹, Salas model², Pasquali model³ and G-score⁴ was 44 months, 32.8 months, 38 months, and 84 months respectively. Notably, G-score with longer median follow-up time was claimed to predict both early and late recurrence in SFTs⁴.

To address this concern, we have updated the follow-up status of these two validation cohorts, as the last follow-up time of these two validation cohorts was June, 2022 when we prepared the manuscript. As a result, the median follow-up time for the FAHSYSU cohort and CHCAMS cohort 1 has been extended to 32.30 months (ranging from 0.0 to 121.20 months) and 36.00 months (ranging from 0.0 to 73.90 months), respectively (**Table 2**). With this updated follow-up, our integrated risk model still demonstrates successful validation in both the FAHSYSU cohort and CHCAMS cohort 1 (**Fig. 4d-e, 5c-f, Supplementary Fig.**

2a-f, 6a-g, 7a-h). Notably, the integrated risk model was also successfully validated in the CHCAMS cohort 2, which boasts a considerably longer median follow-up time of 72.6 months (**Fig. 5g-h**). The successful validation of our integrated risk model in this cohort further substantiates its ability to predict both early and late recurrence/metastasis in SFTs.

Taken together, these adjustments and additional validations in cohorts with varying follow-up durations provide strong evidence supporting the relevance and effectiveness of our prognostic model for SFT patients. We appreciate the reviewer's valuable input and believe that these enhancements have addressed the concerns effectively.

The authors state that their model showed better accuracy and improved efficacy compared to the current WHO classification, mDemicco model, and G-score, and that it represents a significant improvement. Indeed, the C-index and AUC of the integrated model were numerically better than those of the other models. I agree that the proposed model performs well. I am not convinced, however, how large improvement the model represents in clinical practice. Since the application of the model would necessitate additional immunohistochemistry and MTOR mutation analysis, the authors should critically discuss how the application of the model might improve clinical care and therapeutic decisions, and whether this potential improvement justifies the additional investigations needed.

RESPONSE: We extend our sincere gratitude to the reviewer for the meticulous evaluation of our study and for providing valuable insights. We acknowledge the need to address the clinical relevance and implications of our integrated risk model in more depth. Indeed, our integrated risk model, which incorporates various factors including mitotic count, density of Ki-67+ cells, CD163+ cells, and *MTOR* mutation status, has demonstrated superior accuracy in identifying high-risk primary non-CNS and CNS SFTs with NTM, outperforming

established models such as the WHO classification, mDemicco model, and G-score. Additionally, the applicability of our integrated model to both CNS and non-CNS SFTs broadens its clinical utility.

To address the reviewer's important concern, we recognize that the practical implications of our model must be thoroughly evaluated in the context of clinical care and therapeutic decision-making. We offer the following critical discussion of the potential clinical benefits and justifications for the additional investigations required: **1) Enhanced therapeutic decision:** With targeted gene NGS testing for mutation information becoming routine in both academic institutions and commercial entities, the incorporation of *MTOR* mutation status into our model is increasingly feasible. Moreover, the identification of potential therapeutic targets by NGS, such as the *IDH1* p.R132S mutation, and the detection of CD163+ macrophages and PD-L1 expression by IHC, could stimulate further research into new treatments for SFTs, such as combinational immune therapy and targeted therapy. **2) Clinical risk stratification:** Our model provides a more accurate assessment of a patient's risk of tumor progression, enabling improved risk stratification. This information can guide clinicians in deciding the frequency of follow-up assessments and the intensity of surveillance, and optimize healthcare resource allocation. Our model could identify high-risk patients at an earlier stage, allowing for more aggressive treatment strategies that could lead to better outcomes. By accurately identifying low-risk patients, our model helps avoid unnecessary over-treatment, reducing both patient risk and healthcare costs.

In conclusion, while the application of our integrated risk model necessitates additional investigations, we firmly believe that the potential improvements it offers in clinical care and therapeutic decisions are substantial and justifiable. We deeply appreciate the reviewer's invaluable input and have incorporated these discussions into our manuscript to provide a more comprehensive view

of the clinical relevance and implications of our findings in the **Discussion section (Page 20-21, Line 423 – 447)**.

1. Demicco, E.G., *et al.* Risk assessment in solitary fibrous tumors: validation and refinement of a risk stratification model. *Modern pathology : an official journal of the United States and Canadian Academy of Pathology, Inc* **30**, 1433–1442 (2017).
2. Salas, S., *et al.* Prediction of local and metastatic recurrence in solitary fibrous tumor: construction of a risk calculator in a multicenter cohort from the French Sarcoma Group (FSG) database. *Ann Oncol* **28**, 1979–1987 (2017).
3. Pasquali, S., *et al.* Resectable extra-pleural and extra-meningeal solitary fibrous tumours: A multi-centre prognostic study. *Eur J Surg Oncol* **42**, 1064–1070 (2016).
4. Georgiesh, T., Boye, K. & Bjerkehagen, B. A novel risk score to predict early and late recurrence in solitary fibrous tumour. *Histopathology* **77**, 123–132 (2020).

REVIEWERS' COMMENTS

Reviewer #3 (Remarks to the Author):

Thank you for your detailed response to my questions and requests. I believe my comments were properly adressed and have nothing more to add.